# Using Surrogates in Covariate-adjusted Response-adaptive Randomized Experiments with Delayed Outcomes

**Lei Shi**
Division of Biostatistics
University of California, Berkeley
2121 Berkeley Way, Berkeley, CA 94704
`leishi@berkeley.edu`

**Waverly Wei**
Marshall School of Business
University of Southern California
3670 Trousdale Pkwy, Los Angeles, CA 90089
`waverly@marshall.usc.edu`

**Jingshen Wang**
Division of Biostatistics
2121 Berkeley Way, Berkeley, CA 94704
`jingshenwang@berkeley.edu`

## Abstract

Covariate-adjusted response-adaptive randomization (CARA) designs are gaining increasing attention. These designs combine the advantages of randomized experiments with the ability to adaptively revise treatment allocations based on data collected across multiple stages, enhancing estimation efficiency. Yet, CARA designs often assume that primary outcomes are immediately observable, which is not the case in many clinical scenarios where there is a delay in observing primary outcomes. This assumption can lead to significant missingness and inefficient estimation of treatment effects. To tackle this practical challenge, we propose a CARA experimental strategy integrating delayed primary outcomes with immediately observed surrogate outcomes. Surrogate outcomes are intermediate clinical outcomes that are predictive or correlated with the primary outcome of interest. Our design goal is to improve the estimation efficiency of the average treatment effect (ATE) of the primary outcome utilizing surrogate outcomes. From a methodological perspective, our approach offers two benefits: First, we accommodate arm and covariates-dependent delay mechanisms without imposing any parametric modeling assumptions on the distribution of outcomes. Second, when primary outcomes are not fully observed, surrogate outcomes can guide the adaptive treatment allocation rule. From a theoretical standpoint, we prove the semiparametric efficiency bound of estimating ATE under delayed primary outcomes while incorporating surrogate outcomes. We show that the ATE estimator under our proposed design strategy attains this semiparametric efficiency bound and achieves asymptotic normality. Through theoretical investigations and a synthetic HIV study, we show that our design is more efficient than the design without incorporating any surrogate information.

## 1 Introduction

### 1.1 Motivation

In recent years, covariate-adjusted response-adaptive randomization (CARA) designs have become increasingly prominent in clinical research for evaluating the effects of treatment plans or drugs on

38th Conference on Neural Information Processing Systems (NeurIPS 2024).

primary outcomes of interest [43, 36, 49]. Specifically, CARA designs are experiment strategies that adaptively revise treatment allocation based on observed outcomes and covariate information accumulated during multi-stage clinical trials [22]. Inheriting the advantage of traditional clinical trial designs, CARA designs can also help with providing reliable causal evidence to support clinical discoveries. Nevertheless, different from traditional clinical trial designs, CARA designs offer several major benefits. On the one hand, CARA designs enhance the precision of medicine by improving the statistical power to estimate treatment effects. On the other hand, CARA designs allow for adaptive modifications of treatment allocation based on intermediate findings. As highlighted in the recent Food and Drug Administration (FDA)'s adaptive design clinical trials for drugs and biologics guidance for industry [43], the key advantage of adaptive clinical trial designs is their ability to adapt to new information that emerges during the trial. This fundamental characteristic fosters more flexible and efficient clinical trials. In sum, CARA designs can utilize clinical trial data efficiently by performing sequentially adaptive subgroup-specific optimization.

While CARA designs are beneficial, they face a practical challenge when outcomes are observed with delays. Such delays make it impossible to sequentially revise the randomization probability to optimize trial objectives. Delayed outcomes are common in clinical practice, as treatments may take time to show their effects, especially for chronic conditions or diseases with slow progression. For instance, consider an HIV study conducted in Tanzania that investigates the effect of cash incentives on viral load suppression [15]. The primary outcome is the viral load measured six months after starting antiretroviral therapy (ART) because achieving viral suppression (below 1000 copies per mL) is critical for preventing HIV transmission [33, 31]. However, measuring viral load depends on regular clinic visits by study subjects, which are often delayed due to factors such as limited access to care, poverty, or social stigma [19]. These delays lead to significant missingness in primary outcome data, which undermines the efficiency of estimating treatment effects on the primary outcome.

In situations where primary outcomes are delayed, surrogate outcomes–intermediate or substitute biomarkers that can be measured immediately and are predictive of primary outcomes–emerge as a natural remedy [20, 29, 5]. As mentioned in the FDA's surrogate endpoint resources for drug and biologic development, from 2010 to 2012, the FDA granted approval to 45% of new drugs based on a surrogate endpoint [44]. Using surrogate outcomes provides early indicators of a treatment's efficacy, which accelerates the decision-making process and can lead to faster delivery of beneficial treatments to patients. Surrogate outcomes are particularly useful in CARA designs because they enable modifications of treatment allocation without waiting for delayed primary outcomes during the interim stages.

As such, there is a pressing need to develop new CARA designs to answer the following questions: How can both delayed primary outcomes and immediately observed surrogate outcomes be used simultaneously to guide treatment allocation revisions, optimizing the estimation efficiency of the primary outcome's treatment effect? In this manuscript, we propose a multi-stage CARA design with the goal of improving the estimation efficiency of primary outcome treatment effects by utilizing surrogate outcomes in the presence of delayed primary outcomes.

Our contributions can be summarized as follows:

1. From a methodological perspective, we propose a novel CARA design strategy that significantly improves the estimation efficiency of the primary outcome's ATE by incorporating surrogate outcomes. First, our approach uniquely integrates both primary and surrogate outcomes synergistically, unlike traditional clinical trial designs that depend exclusively on one or the other. Second, diverging from some adaptive experimental strategies that incorporate both surrogate and primary outcomes, we explore strategies that tackle the arm-dependent delay mechanism of primary outcomes and optimize the statistical efficiency in the presence of temporary or permanent missingness.

2. From a theoretical perspective, we prove the semiparametric efficiency bound under arm-dependent delayed primary outcomes while incorporating surrogate outcomes for estimating the ATE (Theorem 3). We show that our proposed design strategy converges to the oracle design, which represents the optimal design assuming perfect knowledge of the data-generating distribution. We also prove the statistical validity of our inference procedure by establishing the asymptotic normality of our proposed ATE estimator (Theorem 1). Furthermore, the proposed ATE estimator attains the semiparametric efficiency bound (Theorem 2), which verifies the efficiency gain of adopting the proposed design. Theorem 1 further provides

practical guidance for the general setup of the design in terms of the number of experimentation stages and the portion of enrolled units within each stage to achieve optimality. Lastly, we illustrate that our design largely enhances statistical efficiency compared to the design that only uses delayed primary outcomes.

## 1.2 Related literature

**CARA designs.** When covariate information is available to experimenters, CARA design can utilize both outcome information and covariate information to optimize for the experimental goals [37, 47]. CARA designs originate from response-adaptive randomization designs, yet they further utilize covariate information to assist the design [34, 21, 45]. [22] introduced a family of CARA designs that consider both efficiency and ethics. Generalizations of CARA to incorporate semiparametric estimates are discussed by [49]. Related CARA designs are also explored in studies such as [35, 1, 48]. However, unlike conventional CARA designs that assume immediate observation of outcomes, our approach accommodates delayed outcomes.

**Criteria of surrogacy.** [32] proposes the statistical surrogacy criterion, which states that the observed primary outcome should be conditionally independent of the assigned treatment given the observed value of the post-treatment variable (surrogate outcome). The proposal is further extended by [17] and [9]. [16] argues that [32]'s proposal does not satisfy the causal necessity properties and proposes to study the principal surrogate. [27] introduces the notion of a strong surrogate using a graphical model, which states that treatment serves as an instrument for the effect of the surrogate on the primary outcome. [12] makes a detailed comparison of different notions of surrogacy. The authors propose (strictly) consistent surrogacy to avoid the surrogacy paradox and evaluate several conditions for the criteria to hold. [24] studies criteria of surrogacy based on distributional average causal effects.

**Surrogates in randomized experiments.** Existing literature focuses on using surrogates can be roughly divided into two lines of work. The first line of work utilizes surrogates from a data fusion perspective. [4] proposes to use the surrogate index to estimate long-term causal effects when primary outcomes are missing in experimental data and only observed in observational data. This work relies on the statistical surrogacy assumption. [3] proposes a data combination framework to incorporate surrogate outcomes under the assumption of *latent unconfoundedness*. [23] proposes methods to use surrogate outcomes as proxy variables for persistent confounders. The second line of work uses surrogates as supplements for the primary outcomes. [13] investigates using surrogate data to improve efficiency when inferring parameters from estimating functions. [25] proposes to view surrogates as supplements instead of replacements for the primary outcome and investigates how this proposal can improve the efficiency of treatment effect estimation. [10] proposes to combine primary data and auxiliary data where both datasets are assumed to satisfy ignorability. [2] uses surrogate outcomes to assist with the design of adaptive clinical trials and show their relative and absolute benefits. Diverging from the usage of surrogate outcomes in randomized experiments, we explicitly model the arm and covariates delay mechanisms of primary outcomes while incorporating surrogates.

## 2 Problem setup

Suppose we are designing an adaptive experiment with a total of $T$ stages. Let $X_{it}$ denote the covariates, $S_{it}$ denote the surrogate outcome, and $Y_{it}$ denote the primary outcome for subject $i$ at stage $t = 1, \ldots, T$. Let $A_{it}$ denote the treatment assignment status, where $A_{it} = 1$ corresponds to the treatment arm, while $A_{it} = 0$ corresponds to the control. Following the Neyman-Rubin causal inference framework [30, 38], we let $Y_{it}(a)$ and $S_{it}(a)$ represent the primary potential outcome and surrogate potential outcomes under treatment $a$, where $a \in \{0, 1\}$. Assume that $(Y_{it}(1), Y_{it}(0), S_{it}(1), S_{it}(0))$ are i.i.d. copies of a population tuple $(Y(1), Y(0), S(1), S(0))$. Our parameter of interest is the average treatment effect corresponding to the primary outcome:

$$\tau = \mathbb{E}[Y(1) - Y(0)].$$

We are in a setting where, at the end of Stage $t$, we cannot observe the primary outcome for all subjects due to possible delay in the data collection process, but we can observe the surrogate outcomes for all subjects. The surrogate outcomes can inform the primary outcomes. We assume the primary outcome is observed after $D_{it}$ stages and denote the conditional cumulative distribution function of $D_{it}$ as

$$\rho_a(d|x) = \mathbb{P}(D_{it} \le d | A_{it} = a, X_{it} = x), \quad d = \{0, 1, 2, \ldots, T-1\} \cup \{\infty\}, \ a \in \{0, 1\}.$$

$D_{it}$ represents the number of stages for which an observation is delayed at stage $t$. For example, in a two-stage experiment, $D_{i1} = 0$ means no delay for the $i$-th observation. $D_{i1} = 1$ means that the $i$-th primary outcome will be observed at stage 2. When $D_{i1} = \infty$, we never collect the $i$-th primary outcome back. A primary outcome can only be observed by the end of the experiment if $D_{it} \leq T - t$. Moreover, we allow $D_{it} = \infty$, which means that the primary outcome $Y_{it}$ is missing (censored) at the end of the experiment.

The observed primary outcome can be written as

$$
Y_{it} = \begin{cases} A_{it}Y_{it}(1) + (1 - A_{it})Y_{it}(0), & D_{it} \leq T - t; \\ \text{missing}, & D_{it} > T - t. \end{cases}
$$

The observed surrogate outcome can be written as $S_{it} = A_{it}S_{it}(1) + (1 - A_{it})S_{it}(0)$. Therefore, our observed data has the following structure:

$$
\{(X_{it}, A_{it}, S_{it}, Y_{it}, D_{it})_{i=1}^{n_t}\}_{t=1}^{T}.
$$

In our designs, experimenters can sequentially revise the treatment assignment based on the historical data. We define the treatment assignment probability for subjects in stage $t$ as

$$
e_t = \mathbb{P}(A_{it} = 1 | X_{it}, \boldsymbol{\mathcal{H}}_{t-1}), \quad t = 1, \ldots, T,
$$

where $\boldsymbol{\mathcal{H}}_{t-1} = \{(X_{is}, A_{is}, D_{is}, S_{is}, Y_{is})_{i=1}^{n_s}\}_{s=1}^{t-1}$ is the historical information collected up to Stage $t - 1$. In our manuscript, we consider a multi-stage clinical trial setting where $T < \infty$ and $n_t \to \infty$ for $t = 1, \ldots, T$.

We have the following unconfounded assumption for arm-dependent delayed outcomes: $A \perp\!\!\!\perp (Y(1), Y(0), S(1), S(0)) \mid X$. We also assume the delay is arm and covariates dependent: $D \perp\!\!\!\perp (Y(1), Y(0), S(1), S(0)) \mid X, A$. As we are in the randomized experiment setting, the unconfoundedness assumption holds by design. The delay mechanism says that the delay variable is independent of potential primary outcomes and potential surrogate outcomes when conditional one covariates and treatment arms. This rules out the arrow between the delay and the outcome, which simplifies the technical discussion of the paper. Moreover, such a setup aims to reflect real-world randomized experiments where the delay is mainly due to protocol or structural factors instead of the realized outcome (such as the severity of some diseases). For example, if we are testing the effect of a vitamin supplement on blood vitamin levels and ask patients for regular check-ins, the delay is less likely to depend on the outcome since the outcome does not cause severe symptoms that affect patient check-in.

## 3 Design objective

In this section, we shall formulate our design objective. Our goal of the CARA design is to improve the estimation efficiency of the ATE. Heuristically, we aim to find the optimal treatment allocation that minimizes the semiparametric efficiency bound (SPEB) of estimating the ATE.

The formulation of the design objective relies on the derivation of the SPEB of the ATE under the delayed outcome setting while incorporating surrogate information. In Theorem 3 of the Appendix, we establish the SPEB for estimating $\tau$. If we introduce the following notations on the conditional expectation of the outcomes: $\tau(a, x, s) = \mathbb{E}[Y | A = a, X = x, S = s]$, $\tau(a, x) = \mathbb{E}[\tau(A, X, S) | A = a, X = x]$, our design objective can be formally defined as follows:

$$
\min_{e \in [\delta, 1-\delta]^T} \mathbb{V}_{\text{SPEB}}
$$

$$
= \min_{e \in [\delta, 1-\delta]^T} \left\{ \mathbb{E}\left[\frac{(Y - \tau(1, X, S))^2}{\sum_{t=1}^{T} r_t e_t(X) \rho_1(T - t | X)} + \frac{(Y - \tau(0, X, S))^2}{\sum_{t=1}^{T} r_t (1 - e_t(X)) \rho_0(T - t | X)}\right] \right.
$$

$$
\left. + \mathbb{E}\left[\frac{(\tau(1, X, S) - \tau(1, X))^2}{\sum_{t=1}^{T} r_t e_t(X)} + \frac{(\tau(0, X, S) - \tau(0, X))^2}{\sum_{t=1}^{T} r_t (1 - e_t(X))}\right] \right\}.
$$

The SPEB is a theoretical result derived from the work of [28]. Our design objective says that we aim to find the sequence of optimal treatment allocation that minimizes the semiparametric efficiency bound. Furthermore, we require that the treatment allocation should be bounded by $\delta$ and $1 - \delta$ for

$\delta \in (0, 1/2)$. Note that our design objective depends on the conditional expectation of the primary outcome on covariates and surrogates and the delay mechanism. However, in practical settings, both the conditional expectations and the delay mechanisms are unknown to practitioners. Therefore, we propose to use CARA designs that allow us to adaptively learn the unknown parameters based on sequentially collected data. We shall introduce our proposed design strategy in the following section.

## 4 Proposed CARA design strategy

In this section, we shall formally introduce our proposed design strategy. We consider a delay variable $D$ with finite support: $\mathcal{D} = \{0, 1, \ldots, D^\star\} \cup \{\infty\}$, where $D^\star$ is pre-specified based on domain knowledge. Also, we allow $D$ to take $\infty$, meaning the outcome is permanently missing. For simplicity, we assume that the enrollment proportion at each stage is $r_t = 1/T$, for $t = 1, \ldots, T$, $T < \infty$. We assume both $X$ and $S$ take discrete values to avoid unnecessary technical complexity.

From a high level, our proposal consists of four steps. Step 1 consists of $D^\star + 1$ stages and serves as an exploration step that learns parameters like the delay distribution, outcome model, etc. Step 2 is a policy optimization step after Stage $D^\star + 1$, to compute the optimal allocation with the learned parameters. Step 3 is a policy calibration step from Stage $D^\star + 1$ to $T$, where we calibrate the allocation to match the optimal strategy. Step 4 constructs point and variance estimators based on the data. The full procedure is summarized in Algorithm 1.

---

**Algorithm 1** Surrogate-enhanced adaptive experiment with delayed outcomes

    **Stage 1 (Initialization)**:
1: Enroll $n_1$ participants, and assign treatments with $e_1 = \frac{1}{2}$;
2: Estimate $\widehat{\rho}_a^{(1)}(0|x)$ and let $\widehat{\rho}_a^{(1)}(l|x) = \widehat{\rho}_a^{(1)}(0|x)$, $l = 1, \ldots, T-1$.
    **Stage 2 to $D^* + 1$ (Learn delay mechanism)**:
3: **for** $t \to 2$ to $D^* + 1$ **do**
4:     Enroll $n_t$ subjects and assign treatment with probability $e_t = \frac{1}{2}$;
5:     Update $\widehat{\rho}_a^{(t)}(l|x)$ for $l \leq t-1$.
6: **end for**
7: Obtain the sequence of optimal treatment allocation $\widehat{e}^{(D^*+1)}$ by solving (1).
    **Stage $D^* + 2$ to $T$ (Calibration of treatment allocation)**:
8: **for** $t \to D^* + 2$ to $T$ **do**
9:     Enroll subjects and assign treatments with calibrated probability $\widetilde{e}_t^*$.
10: **end for**
    **After Stage $T$ (Inference)**:
11: Update $\widehat{\tau}$ and $\widehat{\mathbb{V}}$.
12: Construct a two-sided $\alpha$-level confidence interval for $\widehat{\tau}$.

---

Below we take a deeper dive into the details and motivations for each step in Algorithm 1.

**Stage 1**. In line 1 -2, we initialize the experiment by assigning treatments with probability $\widehat{e}_1 = \frac{1}{2}$. After Stage 1, we are able to observe all the surrogate outcomes for $n_1$ subjects and part of the primary outcomes. One can estimate arm-dependent delay with $\rho_a(0|x)$ by counting the portion of the observed sample:

$$\widehat{\rho}_a^{(1)}(0|x) = \frac{\sum_{i=1}^{n_1} \mathbb{1}_{(D_{i1} \leq 0)} \cdot \mathbb{1}_{(X_{i1}=x)} \mathbb{1}_{(A_{i1}=a)}}{\sum_{i=1}^{n_1} \mathbb{1}_{(X_{i1}=x)} \mathbb{1}_{(A_{i1}=a)}}$$

As we only observe outcomes whose delay variable $D_{i1} \leq 0$ at the end of Stage 1, we let $\widehat{\rho}_a^{(1)}(l|x) = \widehat{\rho}_a^{(1)}(0|x)$, $l \geq 1$. We then estimate $\widehat{\tau}^{(1)}(a, x, s)$ and $\widehat{\tau}^{(1)}(a, x)$ by taking a sample average of the observed outcome stratified by $(a, x, s)$ and $(a, x)$, respectively:

$$\widehat{\tau}^{(1)}(a, x, s) = \frac{\sum_{i=1}^{n_1} \mathbb{1}_{(D_{i1} \leq 0)} Y_{i1} \mathbb{1}_{(A_{i1}=a)} \mathbb{1}_{(X_{i1}=x)} \mathbb{1}_{(S_{i1}=s)}}{\sum_{i=1}^{n_1} \mathbb{1}_{(D_{i1} \leq 0)} \mathbb{1}_{(A_{i1}=a)} \mathbb{1}_{(X_{i1}=x)} \mathbb{1}_{(S_{i1}=s)}},$$

$$\widehat{\tau}^{(1)}(a, x) = \frac{\sum_{i=1}^{n_1} \mathbb{1}_{(D_{i1} \leq 0)} Y_{i1} \mathbb{1}_{(A_{i1}=a)} \mathbb{1}_{(X_{i1}=x)}}{\sum_{i=1}^{n_1} \mathbb{1}_{(D_{i1} \leq 0)} \mathbb{1}_{(A_{i1}=a)} \mathbb{1}_{(X_{i1}=x)}}.$$

**Stage 2 to $D^* + 1$.** In line 4 to 5, we keep enrolling subjects and assign subjects to the treatment arm with probability $\widehat{e}_t = 1/2$. Update the estimates of the delay mechanism by counting portions of the observed sample up to the current stage:

$$\widehat{\rho}_a^{(t)}(l|x) = \frac{\sum_{s=1}^{t-1}\sum_{i=1}^{n_s} \mathbb{1}_{(D_{is}\leq l)} \cdot \mathbb{1}_{(X_{is}=x)}\mathbb{1}_{(A_{is}=a)}}{\sum_{s=1}^{t-1}\sum_{i=1}^{n_s} \mathbb{1}_{(X_{is}=x)}\mathbb{1}_{(A_{is}=a)}}, \text{ for } l = 0,\ldots,t-1,$$

and let $\widehat{\rho}_a^{(t)}(l|x) = \widehat{\rho}_a^{(t)}(t-1|x)$ for $l \geq t$. Then update the estimates $\widehat{\tau}^{(t)}(a,x,s)$ and $\widehat{\tau}^{(t)}(a,x)$ as the new stratified sample averages:

$$\widehat{\tau}^{(t)}(a,x,s) = \frac{\sum_{l=1}^{t-1}\sum_{i=1}^{n_l} Y_{il}\mathbb{1}_{(D_{is}\leq t-1)}\mathbb{1}_{(A_{il}=a)}\mathbb{1}_{(X_{il}=x)}\mathbb{1}_{(S_{il}=s)}}{\sum_{l=1}^{t-1}\sum_{i=1}^{n_l} \mathbb{1}_{(D_{is}\leq t-1)}\mathbb{1}_{(A_{il}=a)}\mathbb{1}_{(X_{il}=x)}\mathbb{1}_{(S_{il}=s)}},$$

$$\widehat{\tau}^{(t)}(a,x) = \frac{\sum_{l=1}^{t-1}\sum_{i=1}^{n_l} Y_{il}\mathbb{1}_{(D_{is}\leq t-1)}\mathbb{1}_{(A_{il}=a)}\mathbb{1}_{(X_{il}=x)}}{\sum_{l=1}^{t-1}\sum_{i=1}^{n_l} \mathbb{1}_{(D_{is}\leq t-1)}\mathbb{1}_{(A_{il}=a)}\mathbb{1}_{(X_{il}=x)}}.$$

At the end of Stage $D^* + 1$, we can obtain estimates of all the delay mechanisms: $\widehat{\rho}_a^{(D^*+1)}(0|x),\ldots,\widehat{\rho}_a^{(D^*+1)}(D^*|x)$. We then solve the following optimization problem to find the optimal sequence of $\widetilde{e} = (\widetilde{e}_1,\ldots,\widetilde{e}_T)$:

$$\min_{e\in[\delta,1-\delta]^T} \left\{ \frac{1}{\sum_{s=1}^{D^*+1} n_s} \sum_{s=1}^{D^*+1}\sum_{i=1}^{n_s} \left\{ \frac{(Y_{is}-\widehat{\tau}^{(t)}(1,X_{is},S_{is}))^2}{\sum_{t=1}^T r_t e_t \cdot \widehat{\rho}_1^{(t)}(T-t|X_{it})} + \frac{(Y_{is}-\widehat{\tau}^{(t)}(0,X_{is},S_{is}))^2}{\sum_{t=1}^T r_t(1-e_t)\cdot \widehat{\rho}_0^{(t)}(T-t|X_{it})} \right. \right.$$

$$\left. \left. + \frac{(\widehat{\tau}^{(t)}(1,X_{is},S_{is})-\widehat{\tau}^{(t)}(1,X_{is}))^2}{\sum_{t=1}^T r_t e_t} + \frac{\left(\widehat{\tau}^{(t)}(0,X_{is},S_{is})-\widehat{\tau}^{(t)}(0,X_{is})\right)^2}{\sum_{t=1}^T r_t(1-e_t)} + \lambda_N\|e\|_2^2 \right\} \right\}. \tag{1}$$

In particular, confronted with the possibility that multiple minima could exist for the oracle optimization program, we introduced the penalty term $\lambda_N\|e\|_2^2$, which regularizes the solution. In Theorem 2, we will showcase that, theoretically, if we choose the penalty level to be $\lambda_N = C\sqrt{\log N/N}$, $\widetilde{e}$ will converge in probability to an oracle minima of the population program. In practice, we propose to choose the tuning parameter based on cross-validation, where we split the collected data into multiple folds to tune the best scaling factor $C$ for $\lambda_N = C\sqrt{\log N/N}$.

**Stage $(D^* + 2)$ to $T$.** In line 7 to 9, we calibrate the treatment allocation based on the derived Stage $D^* + 1$ optimal treatment allocation strategy. From Stage $(D^* + 2)$ to Stage $(T - D^*)$, we assign treatments with calibrated probability $\widehat{e}_l$, where

$$\widehat{e}_l = \frac{\sum_{s=1}^{T-D^*} n_s \cdot \widetilde{e}_s - \sum_{s=1}^{D^*+1} n_s \cdot \widehat{e}_s}{\sum_{s=D^*+2}^{T-D^*} n_s}, \text{ for } l = D^*+2,\ldots,T-D^*.$$

and $\widehat{e}_s$ is the actual treatment allocation at each Stage $s$. Then, from Stage $s = (T - D^* + 1)$ to $T$, we assign treatments with probability $\widehat{e}_s = \widetilde{e}_s$, where $s = T - D^* + 1,\ldots,T$. After Stage $T$ (line 11 to 12), we update the delay mechanism and the treatment assignment probability using data collected from the whole experiment.

**After Stage $T$.** We obtain the final ATE estimate $\widehat{\tau}$ based on the efficient influence curve derived in Theorem 3 as

$$\widehat{\tau} = \frac{1}{N}\sum_{t=1}^T\sum_{i=1}^{n_t} \left\{ \frac{A_{it}\mathbf{1}_{(D_{it}\leq T-t)}}{\sum_{t=1}^T r_t \cdot \widehat{e}_t(X_{it})\widehat{\rho}_{t,1}(D_{it}\leq T-t|X)}\left(Y_{it}-\widehat{\tau}(1,X_{it},S_{it})\right) \right.$$

$$- \frac{(1-A_{it})\mathbf{1}_{(D_{it}\leq T-t)}}{\sum_{t=1}^T r_t \cdot (1-\widehat{e}_t(X_{it}))\widehat{\rho}_{t,0}(D_{it}\leq T-t|X)}\left(Y_{it}-\widehat{\tau}(0,X_{it},S_{it})\right)$$

$$+ \frac{A_{it}}{\sum_{t=1}^T r_t \cdot \widehat{e}_t(X_{it})}\left(\widehat{\tau}(1,X_{it},S_{it})-\widehat{\tau}(1,X_{it})\right)$$

$$\left. - \frac{(1-A_{it})}{\sum_{t=1}^T r_t \cdot (1-\widehat{e}_t(X_{it}))}\left(\widehat{\tau}(0,X_{it},S_{it})-\widehat{\tau}(0,X_{it})\right) + \widehat{\tau}(1,X_{it})-\widehat{\tau}(0,X_{it}) \right\},$$

and the variance estimates as

$$\widehat{\mathbb{V}} = \frac{1}{N} \sum_{t=1}^{T} \sum_{i=1}^{n_t} \frac{\mathbb{1}_{(D_{it} \leq T-t)} A_{it} (Y_{it} - \widehat{\tau}(1, X_{it}, S_{it}))^2}{\{\sum_{t=1}^{T} r_t \cdot \widehat{e}_t(X_{it}) \widehat{\rho}_1(T-t|X_{it})\}^2} + \frac{\mathbb{1}_{(D_{it} \leq T-t)} (1 - A_{it})(Y_{it} - \widehat{\tau}(0, X_{it}, S_{it}))^2}{\{\sum_{t=1}^{T} r_t \cdot (1 - \widehat{e}_t(X_{it})) \widehat{\rho}_0(T-t|X_{it})\}^2}$$

$$+ \frac{1}{N} \sum_{t=1}^{T} \sum_{i=1}^{n_t} \frac{A_{it}(\widehat{\tau}(1, X_{it}, S_{it}) - \widehat{\tau}(1, X_{it}))^2}{\{\sum_{t=1}^{T} r_t \cdot \widehat{e}_t(X_{it})\}^2} + \frac{(1 - A_{it})(\widehat{\tau}(0, X_{it}, S_{it}) - \widehat{\tau}(0, X_{it}))^2}{\{\sum_{t=1}^{T} r_t \cdot (1 - \widehat{e}_t(X_{it}))\}^2}$$

$$+ \frac{1}{N} \sum_{t=1}^{T} \sum_{i=1}^{n_t} \left( \widehat{\tau}(1, X_{it}) - \widehat{\tau}(0, X_{it}) - \widehat{\tau} \right)^2.$$

## 5    Theoretical investigation

In this section, we investigate the theoretical properties of our proposed design strategy. In Theorem 1, we demonstrate the asymptotic normality of the ATE estimator with any sequence of propensity scores that are generated sequentially from history and converge to some fixed scores. Furthermore, we show that our design strategy asymptotically converges to the oracle design strategy in Theorem 2, which suggests that the constructed estimator is semi-parametrically efficient.

**Theorem 1** (Asymptotic normality). *Assume the delays are conditionally independent of the surrogate and potential outcomes, given covariates and historical data. Assume the following conditions hold:*

1. *Bounded conditional moments:* $\max\{m_4(0, s, x), m_4(1, s, x)\} \leq M_4$, *where*

$$m_4(a, s, x) = \mathbb{E}\left\{ |Y(a)|^4 \mid S(a) = s, X = x \right\}, a = 0, 1.$$

2. *Convergent scores: there exists a set of propensity scores $\mathfrak{e}$, such that*

$$\max_{t \in [T], x \in \mathcal{X}} |\widehat{e}_t(x) - \mathfrak{e}_t(x)| = o_p(1). \tag{2}$$

*Let $n_t = r_t N$ with $r_t > 0$, $\sum_t r_t = 1$. Then when $N \to \infty$, we have*

$$\sqrt{N}(\widehat{\tau} - \tau) \to \mathcal{N}(0, \mathbb{V}(\mathfrak{e})).$$

*Moreover, the variance estimator satisfies*

$$\widehat{\mathbb{V}}(\mathfrak{e}) - \mathbb{V}(\mathfrak{e}) = O_p(\frac{1}{\sqrt{N}}).$$

Theorem 1 is a general theoretical result that works for *any* sequence of historic-dependent treatment allocation strategy $\widehat{e}$, as long as $\widehat{e}$ converges to an oracle policy in probability asymptotically. For example, one can apply complete randomization through the experiment, which corresponds to $\widehat{e}_{t,\text{cr}} = \mathfrak{e}_{t,\text{cr}} = 1/2$ and establish the asymptotic normality of $\widehat{\tau}$ with variance $\mathbb{V}(\mathfrak{e}_{\text{cr}})$. In particular, if the proposed adaptive designing strategy (Algorithm 1) is implemented, the treatment allocation probability will converge to one set of optimal policies that minimizes the semiparametric efficiency bound (4), which is established in Theorem 2 below.

**Theorem 2** (Convergence of the proposed design strategy). *Let $n_t = r_t N$ with $r_t > 0$, $\sum_t r_t = 1$. Assume the bounded conditional moments hold as in Theorem 1. Assume sufficient enrollment at the first $(T - D^\star)$ (non-delayed) stages:*

$$\sum_{t=1}^{T-D^*} r_t \geq \frac{1}{1-\delta} \cdot \frac{\max\left\{ \sqrt{\sigma_0^2/\rho_{0,\infty} + \overline{\sigma}_0^2}, \sqrt{\sigma_1^2/\rho_{1,\infty} + \overline{\sigma}_1^2} \right\}}{\sqrt{\sigma_1^2/\rho_{1,\infty} + \overline{\sigma}_1^2} + \sqrt{\sigma_0^2/\rho_{0,\infty} + \overline{\sigma}_0^2}}. \tag{3}$$

*Set $\lambda_N = C\sqrt{\log N/N}$. When $N \to \infty$, we have*

*(a) $\widetilde{e} - e^* = o_p(1)$, where $e^* = \lim_{\lambda \to 0} e^*(\lambda)$, and*

$$e^*(\lambda) = \underset{e \in [\delta, 1-\delta]^T}{\arg\min} \mathcal{L}(e) + \lambda \|e\|_2^2.$$

*(b) $\widehat{e} - e^{**} = o_p(1)$, where $e^{**}$ is a minima of $\mathcal{L}(e)$ defined as follows:*

$$e_t^{**} = \begin{cases} 1/2, & t \leq D^* + 1; \\ \frac{\sum_{s=1}^{T-D^*} r_s \cdot e_s^* - \sum_{s=1}^{D^*+1} r_s \cdot 1/2}{\sum_{s=D^*+2}^{T-D^*} r_s}, & D^* + 2 \leq t \leq T - D^*; \\ e_t^*, & t \geq T - D^* + 1. \end{cases}$$

Theorem 2 formally established the convergence property of the propensity scores generated by Algorithm 1. More concretely, Theorem 2 showed that two sets of estimated treatment allocation probability converge to the minima of the population optimal variance: one is the scores $\widetilde{e}$ generated by the optimization program (1), and the other is the sequence of propensity scores $\widehat{e}$ upon calibration which is implemented through the experiment. Therefore, Theorem 1 and Theorem 2 together imply that the point estimator based on the proposed treatment strategy attains the semiparametric efficient bound asymptotically.

Theorem 2 imposes an additional condition (3), which requires a sufficient number of enrollment in the non-delayed stages. This is an important sufficient condition for the optimality of the adaptive allocation strategy, which ensures the optimization program has multiple global minima and enables experimenters to calibrate the propensity scores based on the history to achieve optimality. In practice, Condition (3) can also provide guidance on the design of the experiment to decide the length of the study and the portion of units to recruit in the initial stages.

## 6 Synthetic case study: HIV trial

In this synthetic case study, we calibrate our data-generating process using data collected from an HIV randomized controlled trial (RCT) conducted in Tanzania [15]. In this study, $n = 530$ participants are initially enrolled and randomized to one of the three treatment arms: (1) receive no cash, (2) receive 10,000 TZS, and (3) receive 22,500 TZS. In our case study, we combine the latter two arms and define the control arm $A = 0$ as "receive no cash transfer" and the treatment arm $A = 1$ as "receive any cash transfer." For the treatment arm, the cash incentives were administered six times from May 21, 2018, to May 31, 2019. In our case study, we consider June 1, 2019, as the starting point of our trial because all the delays observed after June 1, 2019, are not due to the delay in cash transfer administration. After all the cash incentives were administered, $n = 511$ subjects were still alive on June 1, 2019, with $n = 177$ subjects in the control arm and $n = 334$ in the treatment arm. As the very last test was taken on June 29, 2021, we consider June 29, 2021, as the last time point where we gathered outcome data.

Motivated by the original HIV trial, we also consider a multi-stage trial where each stage consists of six months, with the primary outcome $Y$ being the viral load measured at the end of a stage. The viral load can be delayed to be measured because HIV patients may miss their scheduled test appointments. Therefore, we consider the World Health Organization (WHO) stage as the surrogate outcome $S \in \{1, 2, 3\}$ because the WHO stage is a clinical indicator of HIV infection progression and it is measured without delay, where $S = 1, 2, 3$ indicates the asymptomatic, mild, and advanced symptoms respectively. We use "sex" as the covariate $X$ in our study, where $X = 1$ indicates male and $X = 0$ indicates female. We further observe that the delay mechanism depends on both the treatment arm and the covariate. We summarize the delay mechanism observed from the real data in Table B. We summarize the expectation and standard deviation of the primary outcome under arm $A$ given $X$ and $S$ in Table 2.

In our synthetic data generation, we generate the surrogate outcome from a multinomial distribution. The delay mechanism is also generated following multinomial distribution using the parameters in Table B. We generate X as $X_{it} \sim \text{Bernoulli}(0.64)$, and the primary outcome variable $Y$ as $Y_{it}|A_{it} = a, X_{it} = x, S_{it} = s \sim \mathcal{N}(\tau(a, x, s))$. The true average treatment effect of the primary outcome is $\tau = 0.04$. Following our proposed design strategy, we set the total number of experimental stages as $T = 2(D^* + 1) = 8$. In this case study, We compare three designs: (1) Our proposed design strategy; (2) the complete randomization design which sets $e_t = 1/2$ for $t = 1, \ldots, T$; (3) CARA design that only utilizes primary outcomes but not surrogate outcomes.

We present the case study results in Figure 1. In terms of bias, Figure 1(A) suggests that the point estimators from all three designs have a vanishing bias as the sample size grows, validating the asymptotic unbiasedness of all strategies. Nevertheless, in terms of variance, Figure 1(B) demonstrates

that our proposed design yields smaller standard deviations and thus higher estimation efficiency for estimating the ATE. The synthetic case study results verify our efficiency comparison in Section 5 and demonstrate the efficiency gain of our proposed design strategy.

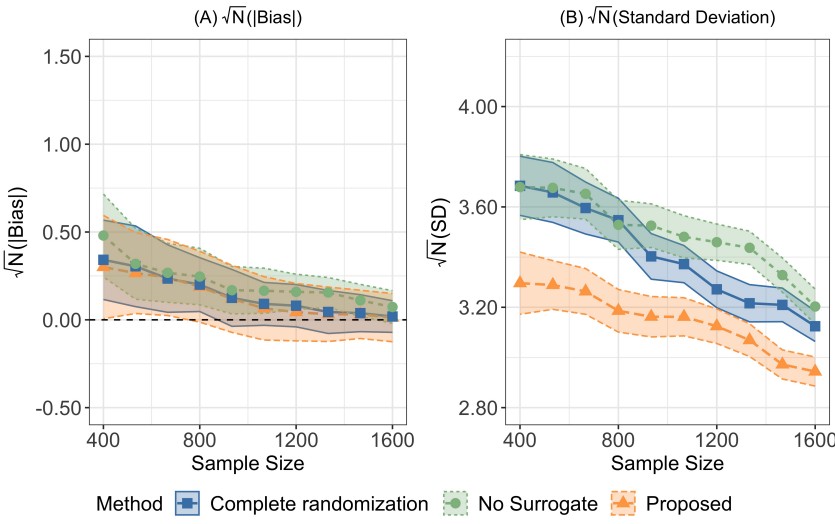

Figure 1: Bias and standard deviation comparison of the three design strategies.

# 7  Discussion

In this work, we proposed a CARA design strategy that improves the efficiency of ATE estimation in the presence of delay by incorporating surrogate outcomes. The strategy leads to a point estimator that achieves semiparametric efficiency under the arm-covariate-dependent delay setting. It also demonstrates efficiency gain compared with the strategy that uses only primary endpoints. There are some future directions to explore. First, we mainly considered an arm-dependent delay setting [18, 41]. A more general setup is to consider the delay mechanism that is also dependent on the outcomes [42]. Another possible framework is to consider outcomes that are also impacted by delays, for which the delay variable serves as a special set of surrogate variables [11]. Moreover, the objective of this work focuses on statistical efficiency gain for ATE estimation. It is also important to incorporate realistic constraints for practical implementation, such as fairness among subgroups [46], safety concerns [6], the discrepancy between algorithmic results and human knowledge [14], etc. More generally, it is also interesting to draw connection to other relevant problems in causal inference and adaptive experiments, such as dealing with many treatment arms [26, 39, 40], targeting other causal parameters [7], incorporating instrumental variables [8], among others. We leave these questions as future research work.

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

# Technical Appendix

## A  Additional results and theoretical proofs

### A.1  Derivation of the semi-parametric efficiency bound

We use $R_i \in \{1, \ldots, T\}$ to record the stage at which the subject $i$ is enrolled, such that $\mathbb{P}(R_i = t) = r_t$, $\sum_{t=1}^{T} r_t = 1$.

**Theorem 3.** *Under the unconfoundedness assumption and the arm-dependent delay assumption:*

*(a) The efficient influence function for estimating $\tau$ is*

$$
\begin{aligned}
&\varphi(X, A, S, Y, R, D) \\
&= \tau(1, X) - \tau(0, X) - \tau \\
&\quad + \frac{A \sum_{t=1}^{T} \mathbb{1}_{(R=t)}}{\sum_{t=1}^{T} r_t e_t(X)} \left(\tau(1, X, S) - \tau(1, X)\right) - \frac{(1-A) \sum_{t=1}^{T} \mathbb{1}_{(R=t)}}{\sum_{t=1}^{T} r_t (1 - e_t(X))} \left(\tau(0, X, S) - \tau(0, X)\right) \\
&\quad + \frac{A \sum_{t=1}^{T} \mathbb{1}_{(R=t)} \mathbb{1}_{(D \leq T-t)}}{\sum_{t=1}^{T} r_t e_t(X) \rho_1(T - t|X)} \left(Y - \tau(1, X, S)\right) - \frac{(1-A) \sum_{t=1}^{T} \mathbb{1}_{(R=t)} \mathbb{1}_{(D \leq T-t)}}{\sum_{t=1}^{T} r_t (1 - e_t(X)) \rho_0(T - t|X)} \left(Y - \tau(0, X, S)\right).
\end{aligned}
$$

*(b) The semiparametric efficiency bond for estimating $\tau$ is*

$$
\begin{aligned}
\mathbb{V}_{\text{SPEB}} &= \mathbb{E}[\left(\tau(1, X) - \tau(0, X) - \tau\right)^2] \\
&\quad + \mathbb{E}\left[\frac{(\tau(1, X, S) - \tau(1, X))^2}{\sum_{t=1}^{T} r_t e_t(X)} + \frac{(\tau(0, X, S) - \tau(0, X))^2}{\sum_{t=1}^{T} r_t (1 - e_t(X))}\right] \\
&\quad + \mathbb{E}\left[\frac{(Y - \tau(1, X, S))^2}{\sum_{t=1}^{T} r_t e_t(X) \rho_1(T - t|X)} + \frac{(Y - \tau(0, X, S))^2}{\sum_{t=1}^{T} r_t (1 - e_t(X)) \rho_0(T - t|X)}\right]. \qquad (4)
\end{aligned}
$$

*Proof.* **(a) Proof of Efficient influence function**

**Step 1.** First, we write out the joint density the observed sample as $f_\eta(x, a, d, v, s, y)$, indexed by $\eta$.

$$
\begin{aligned}
&f_\eta(x, a, d, v, s, y) \\
&= f_{X,\eta}(x) \prod_{t=1}^{T} \Big\{ r_{t,\eta} [e_{t,\eta}(x) g_{S(1)|X,\eta}(s|x) \left(\rho_{a,\eta}(T - t|x) f_{Y(1)|S,X,\eta}(y|s,x)\right)^{\mathbb{1}_{(d \leq T-t)}}] \\
&\qquad (1 - \rho_{a,\eta}(T - t|x))^{\mathbb{1}_{(d > T-t)}}]^a \\
&\qquad \cdot [(1 - e_{t,\eta}(x)) g_{S(0)|X,\eta}(s|x) \left(\rho_{1-a,\eta}(T - t|x) f_{Y(0)|S,X,\eta}(y|s,x)\right)^{\mathbb{1}_{(d \leq T-t)}}] \\
&\qquad (1 - \rho_{1-a,\eta}(T - t|x))^{\mathbb{1}_{(d > T-t)}}]^{1-a} \Big\}^{\mathbb{1}_{(v=t)}}, \\
&= f_{X,\eta}(x) \times \left(r_{t,\eta}^{\sum_{t=1}^{T} \mathbb{1}_{(v=t)}}\right) \\
&\quad \times \prod_{t=1}^{T} (e_{t,\eta}(x))^{a \mathbb{1}_{(v=t)}} (1 - e_{t,\eta}(x))^{(1-a) \mathbb{1}_{(v=t)}} \\
&\quad \times \left(g_{S(1)|X,\eta}(s|x)\right)^{\sum_{t=1}^{T} a \mathbb{1}_{(v=t)}} \left(g_{S(0)|X,\eta}(s|x)\right)^{\sum_{t=1}^{T} (1-a) \mathbb{1}_{(v=t)}} \\
&\quad \times \prod_{t=1}^{T} (\rho_{a,\eta}(T - t|x))^{a \mathbb{1}_{(d \leq T-t)} \mathbb{1}_{v=t}} (1 - \rho_{a,\eta}(T - t|x))^{a \mathbb{1}_{(d > T-t)} \mathbb{1}_{(v=t)}} \\
&\quad \times \prod_{t=1}^{T} (\rho_{1-a,\eta}(T - t|x))^{(1-a) \mathbb{1}_{(d \leq T-t)} \mathbb{1}_{(v=t)}} (1 - \rho_{1-a,\eta}(T - t|x))^{(1-a) \mathbb{1}_{(d > T-t)} \mathbb{1}_{(v=t)}} \\
&\quad \times \left(f_{Y(1)|S,X,\eta}(y|s,x)\right)^{\sum_{t=1}^{T} a \mathbb{1}_{(d \leq T-t)} \mathbb{1}_{(v=t)}}
\end{aligned}
$$

$$\times \left(f_{Y(0)|S,X,\eta}(y|s,x)\right)^{\sum_{t=1}^{T}(1-a)\mathbb{1}_{(d\leq T-t)}\mathbb{1}_{(v=t)}}.$$

**Step 2.** Next, we derive the tangent spaces. We use $\mathfrak{f}(\cdot),\ \mathfrak{g}(\cdot),\ \mathfrak{e}_t(\cdot),\ \mathfrak{p}_t(\cdot)$ to denote $\frac{\partial}{\partial\eta}\log f(\cdot)$, $\frac{\partial}{\partial\eta}\log g(\cdot)$, $\frac{\partial}{\partial\eta}\log e(\cdot)$, and $\frac{\partial}{\partial\eta}\log\rho(\cdot)$.

$$\mathcal{T}_X = \{\mathfrak{f}_X(x) : \mathbb{E}[\mathfrak{f}_X(X_i)] = 0\},$$

$$\mathcal{T}_{Y(1)} = \left\{\sum_{t=1}^{T} a\mathbb{1}_{(d\leq T-t)}\mathbb{1}_{(v=t)}\cdot\mathfrak{f}_1(y|s,x) : \mathbb{E}[\mathfrak{f}_1(Y_i(1)|S_i,X_i)|S_i,X_i]=0\right\},$$

$$\mathcal{T}_{Y(0)} = \left\{\sum_{t=1}^{T} (1-a)\mathbb{1}_{(d\leq T-t)}\mathbb{1}_{(v=t)}\cdot\mathfrak{f}_0(y|s,x) : \mathbb{E}[\mathfrak{f}_0(Y_i(0)|S_i,X_i)|S_i,X_i]=0\right\},$$

$$\mathcal{T}_{S(1)} = \left\{\sum_{t=1}^{T} a\mathbb{1}_{(v=t)}\mathfrak{g}_1(s|x) : \mathbb{E}[\mathfrak{g}_1(S_i(1)|X_i)|X_i]=0\right\}$$

$$\mathcal{T}_{S(0)} = \left\{\sum_{t=1}^{T} (1-a)\mathbb{1}_{(v=t)}\mathfrak{g}_0(s|x) : \mathbb{E}[\mathfrak{g}_0(S_i(0)|X_i)|X_i]=0\right\}$$

$$\mathcal{T}_{A,t} = \Big\{a\mathbb{1}_{(v=t)}\mathfrak{e}_t(x) + (1-a)\mathbb{1}_{(v=t)}\big(1-\mathfrak{e}_t(x)\big)$$
$$= \mathbb{1}_{(v=t)}\Big(\frac{a}{e_t(x)} - \frac{1-a}{1-e_t(x)}\Big)e_t'(x)\Big\},$$

$$\mathcal{T}_{D,a,t} = \Big\{a\mathbb{1}_{(v=t)}\Big(\mathbb{1}_{(d\leq T-t)}\mathfrak{p}_a(T-t|x) + \mathbb{1}_{(d>T-t)}\big(1-\mathfrak{p}_a(T-t|x)\big)\Big)$$
$$= a\mathbb{1}_{(v=t)}\Big(\frac{\mathbb{1}_{(d\leq T-t)}}{\rho_a(T-t|x)} - \frac{\mathbb{1}_{(d>T-t)}}{1-\rho_a(x)}\Big)\rho_a'(T-t|x)\Big\},$$

$$\mathcal{T}_{D,1-a,t} = \Big\{(1-a)\mathbb{1}_{(v=t)}\Big(\mathbb{1}_{(d\leq T-t)}\mathfrak{p}_{1-a}(T-t|x) + \mathbb{1}_{(d>T-t)}\big(1-\mathfrak{p}_{1-a}(T-t|x)\big)\Big)$$
$$= (1-a)\mathbb{1}_{(v=t)}\Big(\frac{\mathbb{1}_{(d\leq T-t)}}{\rho_{1-a}(T-t|x)} - \frac{\mathbb{1}_{(d>T-t)}}{1-\rho_{1-a}(x)}\Big)\rho_{1-a}'(T-t|x)\Big\}.$$

Therefore, the tangent space is

$$\mathcal{T} = \mathcal{T}_X \oplus \mathcal{T}_{Y(1)} \oplus \mathcal{T}_{Y(0)} \oplus \mathcal{T}_{S(1)} \oplus \mathcal{T}_{S(0)} \oplus (\mathcal{T}_{A,1}\oplus\mathcal{T}_{D,A,1}\oplus\mathcal{T}_{1-D,A,1}) \oplus \ldots \oplus (\mathcal{T}_{A,T}\oplus\mathcal{T}_{D,A,T}\oplus\mathcal{T}_{1-D,A,T}).$$

**Step 3.** We consider the estimation of the target parameter $\mathbb{E}[Y(1)]$ via a parametric submodel indexed by $\eta$. Under identifiability assumptions, the target parameter can be written as

$$\mathbb{E}_\eta[Y(1)] = \mathbb{E}_\eta\Big[\mathbb{E}_\eta\big[\mathbb{E}_\eta[Y|A=1,V=t,D\leq T-t,S,X]|X\big]\Big].$$

Taking pathwise derivative of $\mathbb{E}_\eta[Y(1)]$ at $\eta=0$, we have

$$\frac{\partial\mathbb{E}_\eta[Y(1)]}{\partial\eta}\Big|_{\eta=0}$$

$$= \underbrace{\frac{\partial}{\partial\eta}\mathbb{E}_\eta\Big[\mathbb{E}\big[\mathbb{E}[Y|A=1,V=t,D\leq T-t,S,X]|X\big]\Big]\Big|_{\eta=0}}_{(I)}$$

$$+ \underbrace{\mathbb{E}\Big[\frac{\partial}{\partial\eta}\mathbb{E}_\eta\big[\mathbb{E}[Y|A=1,V=t,D\leq T-t,S,X]|X\big]\Big|_{\eta=0}\Big]}_{(II)}$$

$$+ \underbrace{\mathbb{E}\Big[\mathbb{E}\big[\frac{\partial}{\partial\eta}\mathbb{E}_\eta[Y|A=1,V=t,D\leq T-t,S,X]|_{\eta=0}|X\big]\Big]}_{(III)}$$

Let $\mathfrak{s}(X, A, D, V, S, Y)$ denote the score function of $f(x, a, d, v, s, y)$.

$$(I) = \mathbb{E}\left[\tau(1, X)\mathfrak{f}_X(X)\right]$$

$$= \mathbb{E}\left[(\tau(1, X) - \tau_1)\mathfrak{f}_X(X)\right]$$

$$= \mathbb{E}\left[(\tau(1, X) - \tau_1)\mathfrak{s}(X, A, D, V, S, Y)\right],$$

where $\tau(1, X) = \mathbb{E}\left[\mathbb{E}[Y|A = 1, V = t, D \leq T - t, S, X]|X\right]$.

$$(II) = \mathbb{E}\left[\mathbb{E}\left[\tau(1, X, S)\mathfrak{g}(S|X, A = 1)|X\right]\right]$$

$$= \mathbb{E}\left[\mathbb{E}\left[(\tau(1, X, S) - \tau(1, X))\mathfrak{g}(S|X, A = 1)|X\right]\right]$$

$$= \mathbb{E}\left[\mathbb{E}\left[(\tau(1, X, S) - \tau(1, X))\mathfrak{s}(X, A, D, V, S, Y)|X\right]\right]$$

$$= \mathbb{E}\left[\frac{A \cdot \mathbb{1}_{(V=t)}}{e_t(X) \cdot r_t}(\tau(1, X, S) - \tau(1, X))\mathfrak{s}(X, A, D, V, S, Y)\right],$$

where $\tau(1, S, X) = \mathbb{E}[Y|A = 1, V = t, D \leq T - t, S, X]$.

$$(III) = \mathbb{E}\left[\mathbb{E}\left[\mathbb{E}[Y\mathfrak{f}_Y(Y|A = 1, S, X, V, D)|A = 1, V = t, D \leq T - t, S, X]|X\right]\right]$$

$$= \mathbb{E}\left[\mathbb{E}\left[\mathbb{E}[(Y - \tau(1, S, X))\mathfrak{f}_Y(Y|A = 1, S, X, V, D)|A = 1, V = t, D \leq T - t, S, X]|X\right]\right]$$

$$= \mathbb{E}\left[\frac{A \cdot \mathbb{1}_{(V=t)} \cdot \mathbb{1}_{(D\leq T-t)}}{e_t(X)r_t\rho_a(T - t|X)}(Y - \tau(1, S, X))\mathfrak{f}_Y(Y|A = 1, S, X, V, D)\right]$$

$$= \mathbb{E}\left[\frac{A \cdot \mathbb{1}_{(V=t)} \cdot \mathbb{1}_{(D\leq T-t)}}{e_t(X) \cdot r_t \cdot \rho_a(T - t|X)}(Y - \tau(1, S, X))\mathfrak{s}(Y, A, S, X, V, D)\right].$$

As a result,

$$\frac{\partial \mathbb{E}_\eta[Y(1)]}{\partial \eta}\Big|_{\eta=0}$$

$$= \mathbb{E}\left[\left((\tau(1, X) - \tau_1) + \frac{A \cdot \mathbb{1}_{(V=t)}}{e_t(X)r_t}(\tau(1, X, S) - \tau(1, X)) + \frac{A \cdot \mathbb{1}_{(V=t)} \cdot \mathbb{1}_{(D\leq T-t)}}{e_t(X)r_t\rho_a(T - t|X)}(Y - \tau(1, S, X))\right)\right.$$

$$\left.\mathfrak{s}(Y, A, S, X, V, D)\right]$$

$$= \mathbb{E}\left[\left(\varphi_1(X) + \varphi_2(A, S, X, V) + \varphi_3(Y, A, S, X, V, D)\right)\mathfrak{s}(Y, A, S, X, V, D)\right],$$

where

$$\varphi_1(X) = \tau(1, X) - \tau_1,$$

$$\varphi_2(A, S, X, V) = \frac{A \cdot \mathbb{1}_{(V=t)}}{e_t(X)r_t}(\tau(1, X, S) - \tau(1, X)),$$

$$\varphi_3(Y, A, S, X, V, D) = \frac{A \cdot \mathbb{1}_{(V=t)} \cdot \mathbb{1}_{(D \leq T-t)}}{e_t(X) r_t \rho_a(T-t|X)} \big(Y - \tau(1, S, X)\big).$$

**Step 4.** Now we project the score functions to the corresponding tangent spaces. First, we project $\varphi_1(X)$ onto $\mathcal{T}_X$.

$$\mathbb{E}\left[\left(\varphi_1(X) - \mathfrak{f}(X)\right) \times \mathfrak{f}(X)\right] = 0.$$

We obtain

$$\mathfrak{f}(X) = \varphi_1(X).$$

Next, we project $\varphi_2(A, S, X, V)$ onto $\mathcal{T}_{S(1)}$.

$$\mathbb{E}\left[\left(\varphi_2(A, S, X, V) - \left(A \sum_{t=1}^{T} \mathbb{1}_{(V=t)}\right) \mathfrak{f}(S|X)\right) \times \left(A \sum_{t=1}^{T} \mathbb{1}_{(V=t)}\right) \mathfrak{f}(S|X)\right] = 0.$$

$$\mathbb{E}\left[\varphi_2(A, X, V, D) \times \left(A_i \sum_{t=1}^{T} \mathbb{1}_{(V=t)}\right) \mathfrak{f}(S|X)\right]$$

$$= \mathbb{E}\left[\frac{A \cdot \mathbb{1}_{(V=t)}}{e_t(X) r_t} \big(\tau(1, S, X) - \tau(1, X)\big) \times \left(A \sum_{t=1}^{T} \mathbb{1}_{(V=t)}\right) \mathfrak{f}(S|X)\right]$$

$$= \mathbb{E}\left[\mathbb{E}\left[\frac{A \cdot \mathbb{1}_{(V=t)}}{e_t(X) r_t} \big(\tau(1, S, X) - \tau(1, X)\big) \times \left(A \sum_{t=1}^{T} \mathbb{1}_{(V=t)}\right) \mathfrak{f}(S|X)|X\right]\right]$$

$$= \mathbb{E}\left[\big(\tau(1, S, X) - \tau(1, X)\big) \mathfrak{f}(S|X)\right].$$

Then,

$$\mathbb{E}\left[\left(A \sum_{t=1}^{T} \mathbb{1}_{(V=t)}\right) \mathfrak{f}(S|X) \times \left(A \sum_{t=1}^{T} \mathbb{1}_{(V=t)}\right) \mathfrak{f}(S|X)\right]$$

$$= \mathbb{E}\left[\left(A \sum_{t=1}^{T} \mathbb{1}_{(V=t)}\right) \mathfrak{f}(S|X)^2\right]$$

$$= \mathbb{E}\left[\mathbb{E}\left[\left(A \sum_{t=1}^{T} \mathbb{1}_{(V=t)}\right) \mathfrak{f}(S|X)^2|X\right]\right]$$

$$= \mathbb{E}\left[\sum_{t=1}^{T} e_t(X) r_t \mathbb{E}\left[\mathfrak{f}(S|X)^2|X\right]\right].$$

Because

$$\mathbb{E}\left[\big(\tau(1, S, X) - \tau(1, X)\big) \mathfrak{f}(S|X)\right] = \mathbb{E}\left[\sum_{t=1}^{T} e_t(X) r_t \mathbb{E}\left[\mathfrak{f}(S|X)^2|X\right]\right],$$

we obtain

$$\mathfrak{f}(Y|X, S) = \frac{\tau(1, S, X) - \tau(1, X)}{\sum_{t=1}^{T} e_t(X) r_t}.$$

Lastly, we project $\varphi_3(X)$ onto $\mathcal{T}_{Y(1)}$.

$$\mathbb{E}\left[\left(\varphi_3(Y, A, S, X, V, D) - \left(A \sum_{t=1}^{T} \mathbb{1}_{(V=t)} \mathbb{1}_{D \leq T-t}\right) \mathfrak{f}(Y|X, S)\right) \times \left(A \sum_{t=1}^{T} \mathbb{1}_{(V=t)} \mathbb{1}_{D \leq T-t}\right) \mathfrak{f}(Y|X, S)\right] = 0.$$

$$\mathbb{E}\left[\varphi_3(Y, A, S, X, V, D) \times \left(A_i \sum_{t=1}^{T} \mathbb{1}_{(V=t)} \mathbb{1}_{(D \leq T-t)}\right) \mathfrak{f}(Y|X,S)\right]$$

$$= \mathbb{E}\left[\frac{A \cdot \mathbb{1}_{(V=t)} \cdot \mathbb{1}_{(D \leq T-t)}}{e_t(X) r_t \rho_a(T-t|X)} (Y - \tau(1, S, X)) \times \left(A \sum_{t=1}^{T} \mathbb{1}_{(V=t)} \mathbb{1}_{(D \leq T-t)}\right) \mathfrak{f}(Y|X,S)\right]$$

$$= \mathbb{E}\left[\mathbb{E}\left[\frac{A \cdot \mathbb{1}_{(V=t)} \cdot \mathbb{1}_{(D \leq T-t)}}{e_t(X) r_t \rho_a(T-t|X)} (Y - \tau(1, S, X)) \times \left(A \sum_{t=1}^{T} \mathbb{1}_{(V=t)} \mathbb{1}_{D \leq T-t}\right) \mathfrak{f}(Y|X,S)|X\right]\right]$$

$$= \mathbb{E}\left[(Y - \tau(1, S, X)) \mathfrak{f}(Y|X,S)\right].$$

$$\mathbb{E}\left[\left(A \sum_{t=1}^{T} \mathbb{1}_{(V=t)} \mathbb{1}_{D \leq T-t}\right) \mathfrak{f}(Y|X,S)\right) \times \left(A \sum_{t=1}^{T} \mathbb{1}_{(V=t)} \mathbb{1}_{D \leq T-t}\right) \mathfrak{f}(Y|X,S)\right]$$

$$= \mathbb{E}\left[\left(A \sum_{t=1}^{T} \mathbb{1}_{(V=t)} \mathbb{1}_{D \leq T-t}\right) \mathfrak{f}(Y|X,S)^2\right]$$

$$= \mathbb{E}\left[\mathbb{E}\left[\left(A \sum_{t=1}^{T} \mathbb{1}_{(V=t)} \mathbb{1}_{D \leq T-t}\right) \mathfrak{f}(Y|X,S)^2|X\right]\right]$$

$$= \mathbb{E}\left[\sum_{t=1}^{T} e_t(X) r_t \rho_1(T-t|X) \mathbb{E}\left[\mathfrak{f}(Y|X,S)^2|X\right]\right].$$

Because

$$\mathbb{E}\left[(Y - \tau(1, S, X)) \mathfrak{f}(Y|X,S)\right] = \mathbb{E}\left[\sum_{t=1}^{T} e_t(X) r_t \rho_1(T-t|X) \mathbb{E}\left[\mathfrak{f}(Y|X,S)^2|X\right]\right],$$

we obtain

$$\mathfrak{f}(Y|X,S) = \frac{Y - \tau(1, S, X)}{\sum_{t=1}^{T} e_t(X) r_t \rho_1(T-t|X)}$$

In sum, the efficient influence function is

$$\varphi(Y, A, S, X, V, D) = \frac{A \cdot \sum_{t=1}^{T} \mathbb{1}_{(V=t)} \cdot \mathbb{1}_{(D \leq T-t)}}{\sum_{t=1}^{T} e_t(X) r_t \rho_1(T-t|X)} (Y - \tau(1, S, X))$$
$$- \frac{(1-A) \cdot \sum_{t=1}^{T} \mathbb{1}_{(V=t)} \cdot \mathbb{1}_{(D \leq T-t)}}{\sum_{t=1}^{T} (1 - e_t(X)) r_t \rho_0(T-t|X)} (Y - \tau(0, S, X))$$
$$+ \frac{A \cdot \sum_{t=1}^{T} \mathbb{1}_{(V=t)}}{\sum_{t=1}^{T} e_t(X) r_t} (\tau(1, X, S) - \tau(1, X))$$
$$- \frac{(1-A) \cdot \sum_{t=1}^{T} \mathbb{1}_{(V=t)}}{\sum_{t=1}^{T} (1 - e_t(X)) r_t} (\tau(0, X, S) - \tau(0, X))$$
$$+ (\tau(1, X) - \tau(0, X) - \tau).$$

$\square$

### (b) Proof of semiparametric efficiency bound

*Proof.*

$\mathbb{V}[\varphi(X, A, S, Y, R, D)]$

$= \mathbb{V}\left[\tau(1, X) - \tau(0, X) - \tau\right.$

$$+ \frac{A\sum_{t=1}^{T}\mathbb{1}_{(R=t)}}{\sum_{t=1}^{T}r_t e_t(X)}\big(\tau(1,X,S)-\tau(1,X)\big) - \frac{(1-A)\sum_{t=1}^{T}\mathbb{1}_{(R=t)}}{\sum_{t=1}^{T}r_t(1-e_t(X))}\big(\tau(0,X,S)-\tau(0,X)\big)$$

$$+ \frac{A\sum_{t=1}^{T}\mathbb{1}_{(R=t)}\mathbb{1}_{(D\leq T-t)}}{\sum_{t=1}^{T}r_t e_t(X)\rho_1(T-t|X)}\big(Y-\tau(1,X,S)\big) - \frac{(1-A)\sum_{t=1}^{T}\mathbb{1}_{(R=t)}\mathbb{1}_{(D\leq T-t)}}{\sum_{t=1}^{T}r_t(1-e_t(X))\rho_0(T-t|X)}\big(Y-\tau(0,X,S)\big)\Big]$$

$$= \underbrace{\mathbb{V}\Big[\tau(1,X)-\tau(0,X)-\tau\Big]}_{(A)}$$

$$+ \underbrace{\mathbb{V}\left[\frac{A\sum_{t=1}^{T}\mathbb{1}_{(R=t)}}{\sum_{t=1}^{T}r_t e_t(X)}\big(\tau(1,X,S)-\tau(1,X)\big) - \frac{(1-A)\sum_{t=1}^{T}\mathbb{1}_{(R=t)}}{\sum_{t=1}^{T}r_t(1-e_t(X))}\big(\tau(0,X,S)-\tau(0,X)\big)\right]}_{(B)}$$

$$+ \underbrace{\mathbb{V}\left[\frac{A\sum_{t=1}^{T}\mathbb{1}_{(R=t)}\mathbb{1}_{(D\leq T-t)}}{\sum_{t=1}^{T}r_t e_t(X)\rho_1(T-t|X)}\big(Y-\tau(1,X,S)\big) - \frac{(1-A)\sum_{t=1}^{T}\mathbb{1}_{(R=t)}\mathbb{1}_{(D\leq T-t)}}{\sum_{t=1}^{T}r_t(1-e_t(X))\rho_0(T-t|X)}\big(Y-\tau(0,X,S)\big)\right]}_{(C)}$$

$$+ 2\mathrm{Cov}\Bigg[\Big(\tau(1,X)-\tau(0,X)-\tau\Big),$$

$$\underbrace{\Big(\frac{A\sum_{t=1}^{T}\mathbb{1}_{(R=t)}}{\sum_{t=1}^{T}r_t e_t(X)}\big(\tau(1,X,S)-\tau(1,X)\big) - \frac{(1-A)\sum_{t=1}^{T}\mathbb{1}_{(R=t)}}{\sum_{t=1}^{T}r_t(1-e_t(X))}\big(\tau(0,X,S)-\tau(0,X)\big)\Big)\Bigg]}_{(D)}$$

$$+ 2\mathrm{Cov}\Bigg[\Big(\tau(1,X)-\tau(0,X)-\tau\Big),$$

$$\underbrace{\Big(\frac{A\sum_{t=1}^{T}\mathbb{1}_{(R=t)}\mathbb{1}_{(D\leq T-t)}}{\sum_{t=1}^{T}r_t e_t(X)\rho_1(T-t|X)}\big(Y-\tau(1,X,S)\big) - \frac{(1-A)\sum_{t=1}^{T}\mathbb{1}_{(R=t)}\mathbb{1}_{(D\leq T-t)}}{\sum_{t=1}^{T}r_t(1-e_t(X))\rho_0(T-t|X)}\big(Y-\tau(0,X,S)\big)\Big)\Bigg]}_{(E)}$$

$$+ 2\mathrm{Cov}\Bigg[\Big(\frac{A\sum_{t=1}^{T}\mathbb{1}_{(R=t)}}{\sum_{t=1}^{T}r_t e_t(X)}\big(\tau(1,X,S)-\tau(1,X)\big) - \frac{(1-A)\sum_{t=1}^{T}\mathbb{1}_{(R=t)}}{\sum_{t=1}^{T}r_t(1-e_t(X))}\big(\tau(0,X,S)-\tau(0,X)\big)\Big),$$

$$\underbrace{\Big(\frac{A\sum_{t=1}^{T}\mathbb{1}_{(R=t)}\mathbb{1}_{(D\leq T-t)}}{\sum_{t=1}^{T}r_t e_t(X)\rho_1(T-t|X)}\big(Y-\tau(1,X,S)\big) - \frac{(1-A)\sum_{t=1}^{T}\mathbb{1}_{(R=t)}\mathbb{1}_{(D\leq T-t)}}{\sum_{t=1}^{T}r_t(1-e_t(X))\rho_0(T-t|X)}\big(Y-\tau(0,X,S)\big)\Big)\Bigg]}_{(F)}.$$

We work with each term separately.

$$(A) = \mathbb{V}\Big[\tau(1,X)-\tau(0,X)-\tau\Big]$$

$$= \mathbb{E}\Big[\big(\tau(1,X)-\tau(0,X)-\tau\big)^2\Big].$$

$$(B) = \mathbb{V}\left[\frac{A\sum_{t=1}^{T}\mathbb{1}_{(R=t)}}{\sum_{t=1}^{T}r_t e_t(X)}\big(\tau(1,X,S)-\tau(1,X)\big) - \frac{(1-A)\sum_{t=1}^{T}\mathbb{1}_{(R=t)}}{\sum_{t=1}^{T}r_t(1-e_t(X))}\big(\tau(0,X,S)-\tau(0,X)\big)\right]$$

$$= \mathbb{V}\left[\frac{A\sum_{t=1}^{T}\mathbb{1}_{(R=t)}}{\sum_{t=1}^{T}r_t e_t(X)}\big(\tau(1,X,S)-\tau(1,X)\big)\right] + \mathbb{V}\left[\frac{(1-A)\sum_{t=1}^{T}\mathbb{1}_{(R=t)}}{\sum_{t=1}^{T}r_t(1-e_t(X))}\big(\tau(0,X,S)-\tau(0,X)\big)\right]$$

$$- 2\mathrm{Cov}\Bigg[\Big(\frac{A\sum_{t=1}^{T}\mathbb{1}_{(R=t)}}{\sum_{t=1}^{T}r_t e_t(X)}\big(\tau(1,X,S)-\tau(1,X)\big)\Big),$$

$$\Big(\frac{(1-A)\sum_{t=1}^{T}\mathbb{1}_{(R=t)}}{\sum_{t=1}^{T}r_t(1-e_t(X))}\big(\tau(0,X,S)-\tau(0,X)\big)\Big)\Bigg]$$

$$= \mathbb{E}\left[\left(\frac{A\sum_{t=1}^{T}\mathbb{1}_{(R=t)}}{\sum_{t=1}^{T}r_t e_t(X)}\big(\tau(1,X,S)-\tau(1,X)\big)\right)^2\right] + \mathbb{E}\left[\left(\frac{(1-A)\sum_{t=1}^{T}\mathbb{1}_{(R=t)}}{\sum_{t=1}^{T}r_t(1-e_t(X))}\big(\tau(0,X,S)-\tau(0,X)\big)\right)^2\right]$$

$$-2\mathbb{E}\left[\left(\frac{A\sum_{t=1}^{T}\mathbb{1}_{(R=t)}}{\sum_{t=1}^{T}r_t e_t(X)}\big(\tau(1,X,S)-\tau(1,X)\big)\right)\right]\cdot\mathbb{E}\left[\left(\frac{(1-A)\sum_{t=1}^{T}\mathbb{1}_{(R=t)}}{\sum_{t=1}^{T}r_t(1-e_t(X))}\big(\tau(0,X,S)-\tau(0,X)\big)\right)\right]$$

$$= \mathbb{E}\left[\frac{A\sum_{t=1}^{T}\mathbb{1}_{(R=t)}}{\sum_{t=1}^{T}r_t e_t(X)}\big(\tau(1,X,S)-\tau(1,X)\big)^2\right] + \mathbb{E}\left[\frac{(1-A)\sum_{t=1}^{T}\mathbb{1}_{(R=t)}}{\sum_{t=1}^{T}r_t(1-e_t(X))}\big(\tau(0,X,S)-\tau(0,X)\big)^2\right].$$

$$(C) = \mathbb{V}\left[\frac{A\sum_{t=1}^{T}\mathbb{1}_{(R=t)}\mathbb{1}_{(D\le T-t)}}{\sum_{t=1}^{T}r_t e_t(X)\rho_1(T-t|X)}\big(Y-\tau(1,X,S)\big) - \frac{(1-A)\sum_{t=1}^{T}\mathbb{1}_{(R=t)}\mathbb{1}_{(D\le T-t)}}{\sum_{t=1}^{T}r_t(1-e_t(X))\rho_0(T-t|X)}\big(Y-\tau(0,X,S)\big)\right]$$

$$= \mathbb{V}\left[\frac{A\sum_{t=1}^{T}\mathbb{1}_{(R=t)}\mathbb{1}_{(D\le T-t)}}{\sum_{t=1}^{T}r_t e_t(X)\rho_1(T-t|X)}\big(Y-\tau(1,X,S)\big)\right]$$

$$+ \mathbb{V}\left[\frac{(1-A)\sum_{t=1}^{T}\mathbb{1}_{(R=t)}\mathbb{1}_{(D\le T-t)}}{\sum_{t=1}^{T}r_t(1-e_t(X))\rho_0(T-t|X)}\big(Y-\tau(0,X,S)\big)\right]$$

$$- 2\mathrm{Cov}\left[\left(\frac{A\sum_{t=1}^{T}\mathbb{1}_{(R=t)}\mathbb{1}_{(D\le T-t)}}{\sum_{t=1}^{T}r_t e_t(X)\rho_1(T-t|X)}\big(Y-\tau(1,X,S)\big)\right),\right.$$

$$\left.\left(\frac{(1-A)\sum_{t=1}^{T}\mathbb{1}_{(R=t)}\mathbb{1}_{(D\le T-t)}}{\sum_{t=1}^{T}r_t(1-e_t(X))\rho_0(T-t|X)}\big(Y-\tau(0,X,S)\big)\right)\right]$$

$$= \mathbb{E}\left[\left(\frac{A\sum_{t=1}^{T}\mathbb{1}_{(R=t)}\mathbb{1}_{(D\le T-t)}}{\sum_{t=1}^{T}r_t e_t(X)\rho_1(T-t|X)}\big(Y-\tau(1,X,S)\big)\right)^2\right]$$

$$+ \mathbb{E}\left[\left(\frac{(1-A)\sum_{t=1}^{T}\mathbb{1}_{(R=t)}\mathbb{1}_{(D\le T-t)}}{\sum_{t=1}^{T}r_t(1-e_t(X))\rho_0(T-t|X)}\big(Y-\tau(0,X,S)\big)\right)^2\right]$$

$$- 2\mathbb{E}\left[\left(\frac{A\sum_{t=1}^{T}\mathbb{1}_{(R=t)}\mathbb{1}_{(D\le T-t)}}{\sum_{t=1}^{T}r_t e_t(X)\rho_1(T-t|X)}\big(Y-\tau(1,X,S)\big)\right)\right]$$

$$\mathbb{E}\left[\left(\frac{(1-A)\sum_{t=1}^{T}\mathbb{1}_{(R=t)}\mathbb{1}_{(D\le T-t)}}{\sum_{t=1}^{T}r_t(1-e_t(X))\rho_0(T-t|X)}\big(Y-\tau(0,X,S)\big)\right)\right]$$

$$= \mathbb{E}\left[\frac{A\sum_{t=1}^{T}\mathbb{1}_{(R=t)}\mathbb{1}_{(D\le T-t)}}{\sum_{t=1}^{T}r_t e_t(X)\rho_1(T-t|X)}\big(Y-\tau(1,X,S)\big)^2\right]$$

$$+ \mathbb{E}\left[\frac{(1-A)\sum_{t=1}^{T}\mathbb{1}_{(R=t)}\mathbb{1}_{(D\le T-t)}}{\sum_{t=1}^{T}r_t(1-e_t(X))\rho_0(T-t|X)}\big(Y-\tau(0,X,S)\big)^2\right].$$

$$(D) = 2\mathrm{Cov}\left[\big(\tau(1,X)-\tau(0,X)-\tau\big),\right.$$

$$\left.\left(\frac{A\sum_{t=1}^{T}\mathbb{1}_{(R=t)}}{\sum_{t=1}^{T}r_t e_t(X)}\big(\tau(1,X,S)-\tau(1,X)\big) - \frac{(1-A)\sum_{t=1}^{T}\mathbb{1}_{(R=t)}}{\sum_{t=1}^{T}r_t(1-e_t(X))}\big(\tau(0,X,S)-\tau(0,X)\big)\right)\right]$$

$$= \underbrace{2\mathrm{Cov}\left[\big(\tau(1,X)-\tau(0,X)-\tau\big),\left(\frac{A\sum_{t=1}^{T}\mathbb{1}_{(R=t)}}{\sum_{t=1}^{T}r_t e_t(X)}\big(\tau(1,X,S)-\tau(1,X)\big)\right)\right]}_{D_1}$$

$$\underbrace{- 2\mathrm{Cov}\left[\big(\tau(1,X)-\tau(0,X)-\tau\big),\left(\frac{(1-A)\sum_{t=1}^{T}\mathbb{1}_{(R=t)}}{\sum_{t=1}^{T}r_t(1-e_t(X))}\big(\tau(0,X,S)-\tau(0,X)\big)\right)\right]}_{D_2}$$

$$= D_1 - D_2.$$

$$D_1 = 2\mathbb{E}\left[\left(\tau(1,X) - \tau(0,X) - \tau\right)\left(\frac{A\sum_{t=1}^{T}\mathbb{1}_{(R=t)}}{\sum_{t=1}^{T}r_t e_t(X)}\left(\tau(1,X,S) - \tau(1,X)\right)\right)\right]$$

$$- 2\mathbb{E}\left[\left(\tau(1,X) - \tau(0,X) - \tau\right)\right]\mathbb{E}\left[\left(\frac{A\sum_{t=1}^{T}\mathbb{1}_{(R=t)}}{\sum_{t=1}^{T}r_t e_t(X)}\left(\tau(1,X,S) - \tau(1,X)\right)\right)\right]$$

$$= 2\mathbb{E}\left[\left(\tau(1,X) - \tau(0,X) - \tau\right)\left(\frac{A\sum_{t=1}^{T}\mathbb{1}_{(R=t)}}{\sum_{t=1}^{T}r_t e_t(X)}\left(\tau(1,X,S) - \tau(1,X)\right)\right)\right],$$

$$D_2 = 2\mathbb{E}\left[\left(\tau(1,X) - \tau(0,X) - \tau\right)\left(\frac{(1-A)\sum_{t=1}^{T}\mathbb{1}_{(R=t)}}{\sum_{t=1}^{T}r_t(1-e_t(X))}\left(\tau(0,X,S) - \tau(0,X)\right)\right)\right]$$

$$- 2\mathbb{E}\left[\left(\tau(1,X) - \tau(0,X) - \tau\right)\right]\mathbb{E}\left[\left(\frac{(1-A)\sum_{t=1}^{T}\mathbb{1}_{(R=t)}}{\sum_{t=1}^{T}r_t(1-e_t(X))}\left(\tau(0,X,S) - \tau(0,X)\right)\right)\right]$$

$$= 2\mathbb{E}\left[\left(\tau(1,X) - \tau(0,X) - \tau\right)\left(\frac{(1-A)\sum_{t=1}^{T}\mathbb{1}_{(R=t)}}{\sum_{t=1}^{T}r_t(1-e_t(X))}\left(\tau(0,X,S) - \tau(0,X)\right)\right)\right].$$

Therefore,

$$(D) = D_1 - D_2$$

$$= 2\mathbb{E}\left[\left(\tau(1,X) - \tau(0,X) - \tau\right)\left(\frac{A\sum_{t=1}^{T}\mathbb{1}_{(R=t)}}{\sum_{t=1}^{T}r_t e_t(X)}\left(\tau(1,X,S) - \tau(1,X)\right)\right)\right]$$

$$- 2\mathbb{E}\left[\left(\tau(1,X) - \tau(0,X) - \tau\right)\left(\frac{(1-A)\sum_{t=1}^{T}\mathbb{1}_{(R=t)}}{\sum_{t=1}^{T}r_t(1-e_t(X))}\left(\tau(0,X,S) - \tau(0,X)\right)\right)\right]$$

$$= 0.$$

$$(E) = 2\text{Cov}\left[\left(\tau(1,X) - \tau(0,X) - \tau\right),\right.$$

$$\left(\frac{A\sum_{t=1}^{T}\mathbb{1}_{(R=t)}\mathbb{1}_{(D\leq T-t)}}{\sum_{t=1}^{T}r_t e_t(X)\rho_1(T-t|X)}\left(Y - \tau(1,X,S)\right)\right.$$

$$\left.\left.- \frac{(1-A)\sum_{t=1}^{T}\mathbb{1}_{(R=t)}\mathbb{1}_{(D\leq T-t)}}{\sum_{t=1}^{T}r_t(1-e_t(X))\rho_0(T-t|X)}\left(Y - \tau(0,X,S)\right)\right)\right]$$

$$= \underbrace{2\text{Cov}\left[\left(\tau(1,X) - \tau(0,X) - \tau\right), \left(\frac{A\sum_{t=1}^{T}\mathbb{1}_{(R=t)}\mathbb{1}_{(D\leq T-t)}}{\sum_{t=1}^{T}r_t e_t(X)\rho_1(T-t|X)}\left(Y - \tau(1,X,S)\right)\right)\right]}_{E_1}$$

$$\underbrace{- 2\text{Cov}\left[\left(\tau(1,X) - \tau(0,X) - \tau\right), \left(\frac{(1-A)\sum_{t=1}^{T}\mathbb{1}_{(R=t)}\mathbb{1}_{(D\leq T-t)}}{\sum_{t=1}^{T}r_t(1-e_t(X))\rho_0(T-t|X)}\left(Y - \tau(0,X,S)\right)\right)\right]}_{E_2}$$

$$= E_1 - E_2.$$

$$E_1 = 2\mathbb{E}\left[\left(\tau(1,X) - \tau(0,X) - \tau\right)\left(\frac{A\sum_{t=1}^{T}\mathbb{1}_{(R=t)}\mathbb{1}_{(D\leq T-t)}}{\sum_{t=1}^{T}r_t e_t(X)\rho_1(T-t|X)}\left(Y - \tau(1,X,S)\right)\right)\right]$$

$$- 2\mathbb{E}\left[\left(\tau(1,X) - \tau(0,X) - \tau\right)\right]\mathbb{E}\left[\left(\frac{A\sum_{t=1}^{T}\mathbb{1}_{(R=t)}\mathbb{1}_{(D\leq T-t)}}{\sum_{t=1}^{T}r_t e_t(X)\rho_1(T-t|X)}\left(Y - \tau(1,X,S)\right)\right)\right]$$

$$= 0.$$

$$E_2 = 2\mathbb{E}\left[\left(\tau(1,X) - \tau(0,X) - \tau\right)\left(\frac{(1-A)\sum_{t=1}^{T}\mathbb{1}_{(R=t)}\mathbb{1}_{(D\leq T-t)}}{\sum_{t=1}^{T}r_t(1-e_t(X))\rho_0(T-t|X)}\left(Y - \tau(0,X,S)\right)\right)\right]$$

$$- 2\mathbb{E}\left[\left(\tau(1,X) - \tau(0,X) - \tau\right)\right]\mathbb{E}\left[\left(\frac{(1-A)\sum_{t=1}^{T}\mathbb{1}_{(R=t)}\mathbb{1}_{(D\leq T-t)}}{\sum_{t=1}^{T}r_t(1-e_t(X))\rho_0(T-t|X)}\left(Y - \tau(0,X,S)\right)\right)\right]$$

$$= 0.$$

Therefore, $(E) = E_1 - E_2 = 0$.

$$(F) = 2\mathrm{Cov}\left[\left(\frac{A\sum_{t=1}^{T}\mathbb{1}_{(R=t)}}{\sum_{t=1}^{T}r_t e_t(X)}\left(\tau(1,X,S) - \tau(1,X)\right) - \frac{(1-A)\sum_{t=1}^{T}\mathbb{1}_{(R=t)}}{\sum_{t=1}^{T}r_t(1-e_t(X))}\left(\tau(0,X,S) - \tau(0,X)\right)\right),\right.$$

$$\left(\frac{A\sum_{t=1}^{T}\mathbb{1}_{(R=t)}\mathbb{1}_{(D\leq T-t)}}{\sum_{t=1}^{T}r_t e_t(X)\rho_1(T-t|X)}\left(Y - \tau(1,X,S)\right)\right.$$

$$\left.\left. - \frac{(1-A)\sum_{t=1}^{T}\mathbb{1}_{(R=t)}\mathbb{1}_{(D\leq T-t)}}{\sum_{t=1}^{T}r_t(1-e_t(X))\rho_0(T-t|X)}\left(Y - \tau(0,X,S)\right)\right)\right]$$

$$= 2\mathbb{E}\left[\left(\frac{A\sum_{t=1}^{T}\mathbb{1}_{(R=t)}}{\sum_{t=1}^{T}r_t e_t(X)}\left(\tau(1,X,S) - \tau(1,X)\right) - \frac{(1-A)\sum_{t=1}^{T}\mathbb{1}_{(R=t)}}{\sum_{t=1}^{T}r_t(1-e_t(X))}\left(\tau(0,X,S) - \tau(0,X)\right)\right)\right.$$

$$\cdot\left(\frac{A\sum_{t=1}^{T}\mathbb{1}_{(R=t)}\mathbb{1}_{(D\leq T-t)}}{\sum_{t=1}^{T}r_t e_t(X)\rho_1(T-t|X)}\left(Y - \tau(1,X,S)\right)\right.$$

$$\left.\left. - \frac{(1-A)\sum_{t=1}^{T}\mathbb{1}_{(R=t)}\mathbb{1}_{(D\leq T-t)}}{\sum_{t=1}^{T}r_t(1-e_t(X))\rho_0(T-t|X)}\left(Y - \tau(0,X,S)\right)\right)\right]$$

$$- 2\mathbb{E}\left[\left(\frac{A\sum_{t=1}^{T}\mathbb{1}_{(R=t)}}{\sum_{t=1}^{T}r_t e_t(X)}\left(\tau(1,X,S) - \tau(1,X)\right) - \frac{(1-A)\sum_{t=1}^{T}\mathbb{1}_{(R=t)}}{\sum_{t=1}^{T}r_t(1-e_t(X))}\left(\tau(0,X,S) - \tau(0,X)\right)\right)\right.$$

$$\cdot\mathbb{E}\left[\left(\frac{A\sum_{t=1}^{T}\mathbb{1}_{(R=t)}\mathbb{1}_{(D\leq T-t)}}{\sum_{t=1}^{T}r_t e_t(X)\rho_1(T-t|X)}\left(Y - \tau(1,X,S)\right)\right.\right.$$

$$\left.\left.\left. - \frac{(1-A)\sum_{t=1}^{T}\mathbb{1}_{(R=t)}\mathbb{1}_{(D\leq T-t)}}{\sum_{t=1}^{T}r_t(1-e_t(X))\rho_0(T-t|X)}\left(Y - \tau(0,X,S)\right)\right)\right]\right]$$

$$= 2\mathbb{E}\left[\left(\frac{A\sum_{t=1}^{T}\mathbb{1}_{(R=t)}}{\sum_{t=1}^{T}r_t e_t(X)}\left(\tau(1,X,S) - \tau(1,X)\right)\cdot\left(\frac{A\sum_{t=1}^{T}\mathbb{1}_{(R=t)}\mathbb{1}_{(D\leq T-t)}}{\sum_{t=1}^{T}r_t e_t(X)\rho_1(T-t|X)}\left(Y - \tau(1,X,S)\right)\right)\right.\right.$$

$$+ 2\mathbb{E}\left[\left(\frac{(1-A)\sum_{t=1}^{T}\mathbb{1}_{(R=t)}}{\sum_{t=1}^{T}r_t(1-e_t(X))}\left(\tau(0,X,S) - \tau(0,X)\right)\right)\right.$$

$$\left.\left(\frac{(1-A)\sum_{t=1}^{T}\mathbb{1}_{(R=t)}\mathbb{1}_{(D\leq T-t)}}{\sum_{t=1}^{T}r_t(1-e_t(X))\rho_0(T-t|X)}\left(Y - \tau(0,X,S)\right)\right)\right]$$

$$= 0.$$

In sum, terms $(D) = (E) = (F) = 0$, $\mathbb{V}[\varphi(X,A,S,Y,R,D)] = (A) + (B) + (C)$:

$$\mathbb{V}[\varphi(X,A,S,Y,R,D)]$$

$$= \mathbb{E}\left[\left(\tau(1,X) - \tau(0,X) - \tau\right)^2\right]$$

$$+ \mathbb{E}\left[\frac{A\sum_{t=1}^{T}\mathbb{1}_{(R=t)}}{\sum_{t=1}^{T}r_t e_t(X)}\left(\tau(1,X,S) - \tau(1,X)\right)^2\right]$$

$$+ \mathbb{E}\left[\frac{(1-A)\sum_{t=1}^{T}\mathbb{1}_{(R=t)}}{\sum_{t=1}^{T}r_t(1-e_t(X))}\left(\tau(0,X,S) - \tau(0,X)\right)^2\right]$$

$$+ \mathbb{E}\left[\frac{A\sum_{t=1}^{T}\mathbb{1}_{(R=t)}\mathbb{1}_{(D\leq T-t)}}{\sum_{t=1}^{T}r_t e_t(X)\rho_1(T-t|X)}\left(Y-\tau(1,X,S)\right)^2\right]$$

$$+ \mathbb{E}\left[\frac{(1-A)\sum_{t=1}^{T}\mathbb{1}_{(R=t)}\mathbb{1}_{(D\leq T-t)}}{\sum_{t=1}^{T}r_t(1-e_t(X))\rho_0(T-t|X)}\left(Y-\tau(0,X,S)\right)^2\right].$$

$\square$

## A.2 Proof of Theorem 1

*Proof.* Our proposed ATE estimator is

$$\widehat{\tau} = \frac{1}{N}\sum_{t=1}^{T}\sum_{i=1}^{n_t}\left\{\frac{A_{it}\mathbf{1}_{(D_{it}\leq T-t)}}{\sum_{t=1}^{T}r_t\cdot\widehat{e}_t(X_{it})\widehat{\rho}_{t,1}(D_{it}\leq T-t|X)}\left(Y_{it}-\widehat{\tau}(1,X_{it},S_{it})\right)\right.$$

$$-\frac{(1-A_{it})\mathbf{1}_{(D_{it}\leq T-t)}}{\sum_{t=1}^{T}r_t\cdot(1-\widehat{e}_t(X_{it})\widehat{\rho}_{t,0}(D_{it}\leq T-t|X)}\left(Y_{it}-\widehat{\tau}(0,X_{it},S_{it})\right)$$

$$+\frac{A_{it}}{\sum_{t=1}^{T}r_t\cdot\widehat{e}_t(X_{it})}\left(\widehat{\tau}(1,X_{it},S_{it})-\widehat{\tau}(1,X_{it})\right)$$

$$-\frac{(1-A_{it})}{\sum_{t=1}^{T}r_t\cdot(1-\widehat{e}_t(X_{it}))}\left(\widehat{\tau}(0,X_{it},S_{it})-\widehat{\tau}(0,X_{it})\right)$$

$$+\left.\widehat{\tau}(1,X_{it})-\widehat{\tau}(0,X_{it})\right\}.$$

Denote

$$(1)\quad \widehat{\tau}_1 = \frac{1}{N}\sum_{t=1}^{T}\sum_{i=1}^{n_t}\left\{\frac{A_{it}\mathbf{1}_{(D_{it}\leq T-t)}}{\sum_{t=1}^{T}r_t\cdot\widehat{e}_t(X_{it})\widehat{\rho}_{t,1}(D_{it}\leq T-t|X)}\left(Y_{it}-\widehat{\tau}(1,X_{it},S_{it})\right)\right.$$

$$+\left.\frac{A_{it}}{\sum_{t=1}^{T}r_t\cdot\widehat{e}_t(X_{it})}\left(\widehat{\tau}(1,X_{it},S_{it})-\widehat{\tau}(1,X_{it})\right)+\widehat{\tau}(1,X_{it})\right\},$$

$$(2)\quad \widehat{\tau}_0 = \frac{1}{N}\sum_{t=1}^{T}\sum_{i=1}^{n_t}\left\{\frac{(1-A_{it})\mathbf{1}_{(D_{it}\leq T-t)}}{\sum_{t=1}^{T}r_t\cdot(1-\widehat{e}_t(X_{it}))\widehat{\rho}_{t,0}(D_{it}\leq T-t|X)}\left(Y_{it}-\widehat{\tau}(0,X_{it},S_{it})\right)\right.$$

$$+\left.\frac{(1-A_{it})}{\sum_{t=1}^{T}r_t\cdot(1-\widehat{e}_t(X_{it}))}\left(\widehat{\tau}(0,X_{it},S_{it})-\widehat{\tau}(0,X_{it})\right)+\widehat{\tau}(0,X_{it})\right\}.$$

**Step 0. A basic decomposition.**

$$\widehat{\tau}_1 - \tau_1 = \frac{1}{N}\sum_{t=1}^{T}\sum_{i=1}^{n_t}\left\{\underbrace{\frac{A_{it}\mathbf{1}_{(D_{it}\leq T-t)}}{\sum_{t=1}^{T}r_t\cdot\widehat{e}_t(X_{it})\widehat{\rho}_1(D_{it}\leq T-t|X)}\left(Y_{it}-\tau(1,X_{it},S_{it})\right)}_{(A.1)}\right.$$

$$+\underbrace{\left(\frac{A_{it}\mathbf{1}_{(D_{it}\leq T-t)}}{\sum_{t=1}^{T}r_t\cdot\widehat{e}_t(X_{it})\widehat{\rho}_1(D_{it}\leq T-t|X)}-1\right)\left(\tau(1,X_{it},S_{it})-\widehat{\tau}(1,X_{it},S_{it})\right)}_{(A.2)}$$

$$+\underbrace{\frac{A_{it}}{\sum_{t=1}^{T}r_t\cdot\widehat{e}_t(X_{it})}\left(\tau(1,X_{it},S_{it})-\tau(1,X_{it})\right)}_{(B.1)}$$

$$+\underbrace{\left(\frac{A_{it}}{\sum_{t=1}^{T}r_t\cdot\widehat{e}_t(X_{it})}-1\right)\left(\widehat{\tau}(1,X_{it},S_{it})-\tau(1,X_{it},S_{it})-(\widehat{\tau}(1,X_{it})-\tau(1,X_{it}))\right)}_{(B.2)}$$

$$\underbrace{+ \tau(1, X_{it}) - \tau_1}_{(C.1)} \bigg\}.$$

Note that for $\widehat{\tau}_2$, we have a similar decomposition; we omit the presentation for conciseness.

**Step 1. Asymptotics for the nuisance terms.**

   **Step 1.1. Convergence of weighted propensity scores.** Under Condition (2), we prove the following results:

$$\sum_{t=1}^{T} r_t \widehat{e}_t(X_{it}) \widehat{\rho}_1(D_{it} \leq T - t | X) = \sum_{t=1}^{T} r_t \mathfrak{e}_t(X_{it}) \rho_1(D_{it} \leq T - t | X) + o_p(1),$$

$$\sum_{t=1}^{T} r_t(1 - \widehat{e}_t(X_{it})) \widehat{\rho}_0(D_{it} \leq T - t | X) = \sum_{t=1}^{T} r_t(1 - \mathfrak{e}_t(X_{it})) \rho_0(D_{it} \leq T - t | X) + o_p(1),$$

$$\sum_{t=1}^{T} r_t \widehat{e}_t(X_{it}) = \sum_{t=1}^{T} r_t \mathfrak{e}_t(X_{it}) + o_p(1),$$

$$\sum_{t=1}^{T} r_t(1 - \widehat{e}_t(X_{it})) = \sum_{t=1}^{T} r_t(1 - \mathfrak{e}_t(X_{it})) + o_p(1).$$

WLOG, we only prove the first one.

$$\sum_{t=1}^{T} r_t \widehat{e}_t(X_{it}) \widehat{\rho}_1(D_{it} \leq T - t | X)$$

$$= \sum_{t=1}^{T} r_t(\widehat{e}_t(X_{it}) - \mathfrak{e}_t(X_{it})) \widehat{\rho}_1(D_{it} \leq T - t | X) + \sum_{t=1}^{T} r_t \mathfrak{e}_t(X_{it})(\widehat{\rho}_1(D_{it} \leq T - t | X) - \rho_1(D_{it} \leq T - t | X))$$

$$+ \sum_{t=1}^{T} r_t \mathfrak{e}_t(X_{it}) \rho_1(D_{it} \leq T - t | X).$$

By Condition (2), we have

$$\left| \sum_{t=1}^{T} r_t \big(\widehat{e}_t(X_{it}) - \mathfrak{e}_t(X_{it})\big) \widehat{\rho}_1(D_{it} \leq T - t | X) \right|$$

$$\leq \max_{t,x} |\widehat{e}_t(x) - \mathfrak{e}_t(x)| \cdot \sum_{t=1}^{T} r_t \widehat{\rho}_1(D_{it} \leq T - t | X) = o_p(1).$$

Meanwhile,

$$\sum_{t=1}^{T} r_t \widehat{e}_t(x)(\widehat{\rho}_1(D_{it} \leq T - t | x) - \rho_1(D_{it} \leq T - t | x))$$

$$= \sum_{t=1}^{T} \frac{1}{\widehat{n}_t(x)} \sum_{i=1}^{n_t} r_t \widehat{e}_t(x) \left( \frac{\mathbf{1}_{(D_{it} \leq T - t)} A_{it}}{\widehat{e}_t(x)} - \rho_1(D_{it} \leq T - t | x) \right) \mathbf{1}_{(X_{it} = x)}.$$

The summands in the outer layer, indexed by the $t$, form a martingale difference sequence. Hence, its variance is given by

$$\sum_{t=1}^{T} \frac{r_t^2}{n_t} \cdot \mathbb{E} \left\{ \big( \mathbf{1}_{(D_{it} \leq T - t)} A_t - \rho_1(D_{it} \leq T - t | x) \widehat{e}_t(x) \big)^2 \mid X_t = x \right\} p(x) \cdot \mathbb{E} \left\{ \frac{n_t}{\widehat{n}_t(x)} \right\}$$

$$\leq C \cdot \max_t r_t^2 \cdot \sum_{t=1}^{T} \frac{p(x)}{n_t} \cdot \frac{n_t}{p(x)(n_t + 1)}$$

$$\left(\text{where we applied } \mathbb{E}\left\{\frac{1}{1+\text{Bin}(n,p)}\right\} \leq \frac{1}{(1+n)p}\right)$$

$$=O(\frac{1}{n}).$$

Therefore, we have

$$\sum_{t=1}^{T} r_t \widehat{e}_t(X_{it})\widehat{\rho}_1(D_{it} \leq T-t|X) = \sum_{t=1}^{T} r_t \mathfrak{e}_t(X_{it})\rho_1(D_{it} \leq T-t|X) + o_p(1).$$

**Step 1.2. Convergence of the mean functionals.** In this part, we prove the following results:

$$\widehat{\tau}(1,x) - \tau(1,x) = O_p(\frac{1}{\sqrt{N}}),$$

$$\widehat{\tau}(0,x) - \tau(0,x) = O_p(\frac{1}{\sqrt{N}}),$$

$$\widehat{\tau}(1,x,s) - \tau(1,x,s) = O_p(\frac{1}{\sqrt{N}}),$$

$$\widehat{\tau}(0,x,s) - \tau(0,x,s) = O_p(\frac{1}{\sqrt{N}}).$$

WLOG, we check the first result.

$$\widehat{\tau}(1,x) - \tau(1,x) = \frac{N^{-1}\sum_{t=1}^{T}\sum_{i=1}^{n_t}(Y_{it}(1)-\tau(1,x))A_{it}\mathbf{1}_{(D_{it}\leq T-t)}\mathbf{1}_{(X_{it}=x)}}{N^{-1}\sum_{t=1}^{T}\sum_{i=1}^{n_t}A_{it}\mathbf{1}_{(D_{it}\leq T-t)}\mathbf{1}_{(X_{it}=x)}}.$$

The numerator forms a martingale difference triangular array:

$$W_{it}^{(n)} = (Y_{it}(1)-\tau(1,x))A_{it}\mathbf{1}_{(D_{it}\leq T-t)}\mathbf{1}_{(X_{it}=x)}, \quad \mathcal{F}_{it}^{(n)} = \sigma\{(Y_{js}(\cdot),S_{js}(\cdot),X_{js},A_{js},D_{js})_{s\leq t,j\leq i}\}.$$

For convenience we denote $\mathcal{F}_{0s}^{(n)} = \mathcal{F}_{n_{s-1}(s-1)}^{(n)}$, then we can verify that

$$\mathbb{E}\left\{W_{it}^{(n)} \mid \mathcal{F}_{(i-1)t}^{(n)}\right\} = 0, \text{ for } 1 \leq t \leq T, 1 \leq i \leq n_t.$$

The variance of the numerator is then given by

$$\frac{1}{N}\sum_{t=1}^{T} r_t \sigma^2(1,x)\mathbb{E}\left\{\widehat{e}_t(x)\right\}\rho_1^\star(T-t \mid x)p(x)$$

$$=\frac{\sigma^2(1,x)p(x)\sum_{t=1}^{T} r_t \mathfrak{e}_t(x)\rho_1^\star(T-t \mid x) + o(1)}{N}.$$

Therefore,

$$N^{-1}\sum_{t=1}^{T}\sum_{i=1}^{n_t}(Y_{it}(1)-\tau(1,x))A_{it}\mathbf{1}_{(D_{it}\leq T-t)}\mathbf{1}_{(X_{it}=x)} = O_p(\frac{1}{\sqrt{N}}).$$

For the denominator, similarly, we can show that

$$N^{-1}\sum_{t=1}^{T}\sum_{i=1}^{n_t}A_{it}\mathbf{1}_{(D_{it}\leq T-t)}\mathbf{1}_{(X_{it}=x)} = p(x)\sum_{t=1}^{T} r_t \mathfrak{e}_t(x)\rho_1^\star(T-t \mid x) + O_p(\frac{1}{\sqrt{N}}).$$

Therefore,

$$\widehat{\tau}(1,x) - \tau(1,x) = O_p(\frac{1}{\sqrt{N}}).$$

**Step 1.3. Concentration of propensity weights.**

We prove that

$$\frac{1}{N}\sum_{t=1}^{T}\sum_{i=1}^{n_t}\mathbf{1}_{(X_{it}=x)}\left\{\frac{\mathbf{1}_{(D_{it}\leq T-t)}A_{it}}{\sum_{t=1}^{T}r_t\widehat{e}_t(x)\widehat{\rho}_1(T-t|x)}-1\right\}=O_p(\frac{1}{\sqrt{N}}),$$

$$\frac{1}{N}\sum_{t=1}^{T}\sum_{i=1}^{n_t}\mathbf{1}_{(X_{it}=x)}\left\{\frac{\mathbf{1}_{(D_{it}\leq T-t)}(1-A_{it})}{\sum_{t=1}^{T}r_t(1-\widehat{e}_t(x))\widehat{\rho}_0(T-t|x)}-1\right\}=O_p(\frac{1}{\sqrt{N}}),$$

$$\frac{1}{N}\sum_{t=1}^{T}\sum_{i=1}^{n_t}\mathbf{1}_{(X_{it}=x)}\left(\frac{A_{it}}{\sum_{t=1}^{T}r_t\cdot\widehat{e}_t(x)}-1\right)=O_p(\frac{1}{\sqrt{N}}),$$

$$\frac{1}{N}\sum_{t=1}^{T}\sum_{i=1}^{n_t}\mathbf{1}_{(X_{it}=x)}\left(\frac{1-A_{it}}{\sum_{t=1}^{T}r_t\cdot(1-\widehat{e}_t(x))}-1\right)=O_p(\frac{1}{\sqrt{N}}).$$

WLOG, we prove the first result. We can compute

$$\frac{1}{N}\sum_{t=1}^{T}\sum_{i=1}^{n_t}\mathbf{1}_{(X_{it}=x)}\left\{\frac{\mathbf{1}_{(D_{it}\leq T-t)}A_{it}}{\sum_{t=1}^{T}r_t\widehat{e}_t(x)\widehat{\rho}_1(T-t|x)}-1\right\}$$

$$=\frac{\sum_{t=1}^{T}r_t\mathfrak{e}_t(x)\rho_1^\star(T-t\mid x)p(x)+O_p(\frac{1}{\sqrt{N}})}{\sum_{t=1}^{T}r_t\mathfrak{e}_t(x)\rho_1^\star(T-t\mid x)+O_p(\frac{1}{\sqrt{N}})}-\left(p(x)+O_p(\frac{1}{\sqrt{N}})\right)$$

$$=O_p(\frac{1}{\sqrt{N}}).$$

**Step 2. Approximate the main terms.** We prove that the sum of $(A.1)$ and $(B.1)$ can be approximated by the following:

$$\frac{1}{N}\sum_{t=1}^{T}\sum_{i=1}^{n_t}\frac{A_{it}\mathbf{1}_{(D_{it}\leq T-t)}}{\sum_{t=1}^{T}r_t\cdot\widehat{e}_t(X_{it})\widehat{\rho}_1(D_{it}\leq T-t|X_{it})}\left(Y_{it}-\tau(1,X_{it},S_{it})\right)$$

$$=\frac{1}{N}\sum_{t=1}^{T}\sum_{i=1}^{n_t}\frac{A_{it}\mathbf{1}_{(D_{it}\leq T-t)}}{\sum_{t=1}^{T}r_t\cdot\mathfrak{e}_t(X_{it})\rho_1(D_{it}\leq T-t|X_{it})}\left(Y_{it}-\tau(1,X_{it},S_{it})\right)+o_p(\frac{1}{\sqrt{N}}),$$

and

$$\frac{1}{N}\sum_{t=1}^{T}\sum_{i=1}^{n_t}\frac{A_{it}}{\sum_{t=1}^{T}r_t\cdot\widehat{e}_t(X_{it})}\left(\tau(1,X_{it},S_{it})-\tau(1,X_{it})\right)$$

$$=\frac{1}{N}\sum_{t=1}^{T}\sum_{i=1}^{n_t}\frac{A_{it}}{\sum_{t=1}^{T}r_t\cdot\mathfrak{e}_t(X_{it})}\left(\tau(1,X_{it},S_{it})-\tau(1,X_{it})\right)+o_p(\frac{1}{\sqrt{N}}).$$

The first approximation can be done by simply noticing that

$$\frac{1}{N}\sum_{t=1}^{T}\sum_{i=1}^{n_t}A_{it}\mathbf{1}_{(X_{it}=x,S_{it}=s)}\mathbf{1}_{(D_{it}\leq T-t)}\left(Y_{it}-\tau(1,x,s)\right)\cdot$$

$$\left(\frac{1}{\sum_{t=1}^{T}r_t\cdot\widehat{e}_t(x)\widehat{\rho}_1(D_{it}\leq T-t|X_{it})}-\frac{1}{\sum_{t=1}^{T}r_t\cdot\mathfrak{e}_t(x)\widehat{\rho}_1(D_{it}\leq T-t|X_{it})}\right)=O_p(\frac{1}{\sqrt{N}})\cdot o_p(1)=o_p(\frac{1}{\sqrt{N}}).$$

The second approximation can be done similarly.

**Step 3. Compute the variance of the main part.**

By Step 0-2, we have the following approximation for $\widehat{\tau}-\tau$:

$$\widehat{\tau}-\tau=\widehat{\tau}_1-\tau_1-(\widehat{\tau}_0-\tau_0)$$

$$=\widehat{\tau}_{1,\mathrm{apr}}-\tau_1-(\widehat{\tau}_{0,\mathrm{apr}}-\tau_0)+o_p(\frac{1}{\sqrt{N}}),$$

where

$$\widehat{\tau}_{1,\text{apr}} = \frac{1}{N} \sum_{t=1}^{T} \sum_{i=1}^{n_t} \left\{ \frac{A_{it}\mathbf{1}_{(D_{it} \leq T-t)}}{\sum_{t=1}^{T} r_t \cdot \mathfrak{e}_t(X_{it})\rho_1(D \leq T-t|X_{it})} \left(Y_{it} - \tau(1, X_{it}, S_{it})\right) \right.$$

$$\left. + \frac{A_{it}}{\sum_{t=1}^{T} r_t \cdot \mathfrak{e}_t(X_{it})} \left(\tau(1, X_{it}, S_{it}) - \tau(1, X_{it})\right) + \tau(1, X_{it}) \right\},$$

$$\widehat{\tau}_{0,\text{apr}} = \frac{1}{N} \sum_{t=1}^{T} \sum_{i=1}^{n_t} \left\{ \frac{(1-A_{it})\mathbf{1}_{(D_{it} \leq T-t)}}{\sum_{t=1}^{T} r_t \cdot (1-\mathfrak{e}_t(X_{it}))\rho_0(D_{it} \leq T-t|X_{it})} \left(Y_{it} - \tau(0, X_{it}, S_{it})\right) \right.$$

$$\left. + \frac{(1-A_{it})}{\sum_{t=1}^{T} r_t \cdot (1-\mathfrak{e}_t(X_{it}))} \left(\tau(0, X_{it}, S_{it}) - \tau(0, X_{it})\right) + \tau(0, X_{it}) \right\}.$$

We can verify that the variance of $\widehat{\tau}_{1,\text{apr}} - \widehat{\tau}_{0,\text{apr}}$ is given by the following form:

$$\mathbb{V}(\widehat{\tau}_{\text{apr}}) = \frac{1}{N} \cdot \left\{ \mathbb{E}\left\{ \frac{\sigma^2(1, X, S)}{\sum_{t=1}^{T} r_t \cdot \mathfrak{e}_t(X)\rho_1(D \leq T-t|X)} + \frac{\sigma^2(0, X, S)}{\sum_{t=1}^{T} r_t \cdot (1-\mathfrak{e}_t(X))\rho_0(D \leq T-t|X)} \right. \right.$$

$$\left. \left. + \frac{\overline{\sigma}^2(1, X)}{\sum_{t=1}^{T} r_t \cdot \mathfrak{e}_t(X)} + \frac{\overline{\sigma}^2(0, X)}{\sum_{t=1}^{T} r_t \cdot (1-\mathfrak{e}_t(X))} + (\tau(1, X) - \tau(0, X) - \tau)^2 \right\} \right\} + o(\frac{1}{N}).$$

To sum up,

$$\mathbb{V}(\widehat{\tau}_{\text{apr}}) = \frac{1}{N}\mathbb{V}(\mathfrak{e}) + o(\frac{1}{N}).$$

**Step 4. Prove the asymptotic normality of the approximation $\widehat{\tau}_{\text{apr}}$ upon standardization.**

We check that the Lyapunov condition holds:

$$\frac{1}{N^2\mathbb{V}(\mathfrak{e})^2} \sum_{t=1}^{T} \sum_{i=1}^{n_t} \mathbb{E}\left\{ \left| \frac{A_{it}\mathbf{1}_{(D_{it} \leq T-t)}}{\sum_{t=1}^{T} r_t \cdot \mathfrak{e}_t(X_{it})\rho_1(D \leq T-t|X_{it})} \left(Y_{it} - \tau(1, X_{it}, S_{it})\right) \right|^4 \right\}$$

$$\leq \frac{CM_4}{N\mathbb{V}(\mathfrak{e})^2} \mathbb{E}\left\{ \frac{1}{\{\sum_{t=1}^{T} r_t \cdot \mathfrak{e}_t(X_{it})\rho_1(D \leq T-t|X_{it})\}^3} \right\} \leq \frac{CM_4}{N\mathbb{V}(\mathfrak{e})^2\delta^3}.$$

Similarly, we have

$$\frac{1}{N^2\mathbb{V}(\mathfrak{e})^2} \sum_{t=1}^{T} \sum_{i=1}^{n_t} \mathbb{E}\left\{ \left| \frac{(1-A_{it})\mathbf{1}_{(D_{it} \leq T-t)}}{\sum_{t=1}^{T} r_t \cdot (1-\mathfrak{e}_t(X_{it}))\rho_0(D \leq T-t|X_{it})} \left(Y_{it} - \tau(0, X_{it}, S_{it})\right) \right|^4 \right\}$$

$$\leq \frac{CM_4}{N\mathbb{V}(\mathfrak{e})^2} \mathbb{E}\left\{ \frac{1}{\{\sum_{t=1}^{T} r_t \cdot (1-\mathfrak{e}_t(X_{it}))\rho_0(D \leq T-t|X_{it})\}^3} \right\} \leq \frac{CM_4}{N\mathbb{V}(\mathfrak{e})^2\delta^3},$$

and

$$\frac{1}{N^2\mathbb{V}(\mathfrak{e})^2} \sum_{t=1}^{T} \sum_{i=1}^{n_t} \mathbb{E}\left\{ \left| \frac{A_{it}}{\sum_{t=1}^{T} r_t \cdot \mathfrak{e}_t(X_{it})} \left(\tau(1, X_{it}, S_{it}) - \tau(1, X_{it})\right) \right|^4 \right\} \leq \frac{CM_4}{N\mathbb{V}(\mathfrak{e})^2\delta^3},$$

$$\frac{1}{N^2\mathbb{V}(\mathfrak{e})^2} \sum_{t=1}^{T} \sum_{i=1}^{n_t} \mathbb{E}\left\{ \left| \frac{1-A_{it}}{\sum_{t=1}^{T} r_t \cdot (1-\mathfrak{e}_t(X_{it}))} \left(\tau(0, X_{it}, S_{it}) - \tau(0, X_{it})\right) \right|^4 \right\} \leq \frac{CM_4}{N\mathbb{V}(\mathfrak{e})^2\delta^3},$$

$$\frac{1}{N^2\mathbb{V}(\mathfrak{e})^2} \sum_{t=1}^{T} \sum_{i=1}^{n_t} \mathbb{E}\left\{ |\tau(1, X_{it}) - \tau(0, X_{it}) - \tau|^4 \right\} \leq \frac{CM_4}{N\mathbb{V}(\mathfrak{e})^2}.$$

Therefore, we have proved that the Lyapunov condition holds and

$$\sqrt{N}(\widehat{\tau} - \tau) \rightsquigarrow N(0, \mathbb{V}(\mathfrak{e})).$$

**Step 5. Prove the convergence of the variance estimator.**

First, we can prove that

$$\frac{1}{N} \sum_{t=1}^{T} \sum_{i=1}^{n_t} \frac{\mathbf{1}_{(D_{it} \leq T-t)} A_{it} (Y_{it} - \widehat{\tau}(1, X_{it}, S_{it}))^2}{\{\sum_{t=1}^{T} r_t \cdot \widehat{e}_t(X_{it}) \widehat{\rho}_1(D_{it} \leq T - t | X_{it})\}^2}$$

$$= \frac{1}{N} \sum_{t=1}^{T} \sum_{i=1}^{n_t} \frac{\mathbf{1}_{(D_{it} \leq T-t)} A_{it} (Y_{it} - \tau(1, X_{it}, S_{it}) - (\widehat{\tau}(1, X_{it}, S_{it}) - \tau(1, X_{it}, S_{it})))^2}{\{\sum_{t=1}^{T} r_t \cdot \widehat{e}_t(X_{it}) \widehat{\rho}_1(D_{it} \leq T - t | X_{it})\}^2}$$

$$= \frac{1}{N} \sum_{t=1}^{T} \sum_{i=1}^{n_t} \frac{\mathbf{1}_{(D_{it} \leq T-t)} A_{it} (Y_{it} - \tau(1, X_{it}, S_{it}))^2}{\widehat{\pi}_{1T}(X_{it})^2}$$

$$+ \frac{1}{N} \sum_{t=1}^{T} \sum_{i=1}^{n_t} \frac{\mathbf{1}_{(D_{it} \leq T-t)} A_{it} (\widehat{\tau}(1, X_{it}, S_{it}) - \tau(1, X_{it}, S_{it}))^2}{\{\sum_{t=1}^{T} r_t \cdot \widehat{e}_t(X_{it}) \widehat{\rho}_1(D_{it} \leq T - t | X_{it})\}^2}$$

$$- \frac{2}{N} \sum_{t=1}^{T} \sum_{i=1}^{n_t} \frac{\mathbf{1}_{(D_{it} \leq T-t)} A_{it} (Y_{it} - \tau(1, X_{it}, S_{it}))(\widehat{\tau}(1, X_{it}, S_{it}) - \tau(1, X_{it}, S_{it}))}{\widehat{\pi}_{1T}(X_{it})^2}$$

$$= \mathbb{E} \left\{ \frac{\sigma^2(1, X, S)}{\sum_{t=1}^{T} r_t \cdot \mathfrak{e}_t(X) \rho_1(D \leq T - t | X)} \right\} + O_p(\frac{1}{\sqrt{N}}).$$

The results follow from the martingale structure and the approximation given in Steps 1.1 and 1.2.

Analogously, with similar ways of decomposition, we can prove that

$$\frac{1}{N} \sum_{t=1}^{T} \sum_{i=1}^{n_t} \frac{\mathbf{1}_{(D_{it} \leq T-t)} (1 - A_{it})(Y_{it} - \widehat{\tau}(0, X_{it}, S_{it}))^2}{\{\sum_{t=1}^{T} r_t \cdot (1 - \widehat{e}_t(X_{it})) \widehat{\rho}_0(D_{it} \leq T - t | X_{it})\}^2}$$

$$= \mathbb{E} \left\{ \frac{\sigma^2(0, X, S)}{\sum_{t=1}^{T} r_t \cdot (1 - \mathfrak{e}_t(X)) \rho_0(D \leq T - t | X)} \right\} + O_p(\frac{1}{\sqrt{N}}),$$

$$\frac{1}{N} \sum_{t=1}^{T} \sum_{i=1}^{n_t} \frac{\mathbf{1}_{(D_{it} \leq T-t)} A_{it} (\widehat{\tau}(1, X_{it}, S_{it}) - \widehat{\tau}(1, X_{it}))^2}{\{\sum_{t=1}^{T} r_t \cdot \widehat{e}_t(X_{it})\}^2}$$

$$= \mathbb{E} \left\{ \frac{\overline{\sigma}^2(1, X)}{\sum_{t=1}^{T} r_t \mathfrak{e}_t(X)} \right\} + O_p(\frac{1}{\sqrt{N}}),$$

$$\frac{1}{N} \sum_{t=1}^{T} \sum_{i=1}^{n_t} \frac{\mathbf{1}_{(D_{it} \leq T-t)} (1 - A_{it})(\widehat{\tau}(0, X_{it}, S_{it}) - \widehat{\tau}(0, X_{it}))^2}{\{\sum_{t=1}^{T} r_t \cdot (1 - \widehat{e}_t(X_{it}))\}^2}$$

$$= \mathbb{E} \left\{ \frac{\overline{\sigma}^2(0, X)}{\sum_{t=1}^{T} r_t \cdot (1 - \mathfrak{e}_t(X))} \right\} + O_p(\frac{1}{\sqrt{N}}),$$

Moreover, we have

$$\frac{1}{n} \sum_{t=1}^{T} \sum_{i=1}^{n_t} (\widehat{\tau}(1, X_{it}) - \widehat{\tau}(0, X_{it}) - (\widehat{\tau}_{1,\text{OR}} - \widehat{\tau}_{0,\text{OR}}))^2$$

$$= \frac{1}{n} \sum_{t=1}^{T} \sum_{i=1}^{n_t} (\widehat{\tau}(1, X_{it}) - \widehat{\tau}(0, X_{it}) - (\tau(1, X_{it}) - \tau(0, X_{it})))^2$$

$$+ \frac{1}{n} \sum_{t=1}^{T} \sum_{i=1}^{n_t} (\tau(1, X_{it}) - \tau(0, X_{it}) - (\tau_1 - \tau_0))^2$$

$$+\frac{2}{n}\sum_{t=1}^{T}\sum_{i=1}^{n_t}(\tau(1,X_{it})-\tau(0,X_{it})-(\tau_1-\tau_0))(\widehat{\tau}(1,X_{it})-\widehat{\tau}(0,X_{it})-(\tau(1,X_{it})-\tau(0,X_{it})))$$

$$-((\widehat{\tau}_{1,\text{OR}}-\widehat{\tau}_{0,\text{OR}})-(\mu_1-\mu_0))^2$$

$$=\mathbb{E}\left\{(\tau(1,X)-\tau(0,X)-\tau)^2\right\}+O_p(\frac{1}{\sqrt{N}}).$$

Summarizing the above parts, we can conclude

$$\widehat{\mathbb{V}}(\mathfrak{e})-\mathbb{V}(\mathfrak{e})=O_p(\frac{1}{\sqrt{N}}).$$

□

### A.3 Proof of Theorem 2

*Proof.* The proof is delivered in two steps:

**Step 1. A sufficient condition for the existence of non-trivial oracle propensity scores in the non-delay stages.**

For the oracle optimization problem 3, by considering covariates $X$ and $S$ with finite discrete support values, we are essentially solving the following program:

$$\min_{e}\frac{\sigma^2(1,x,s)}{\sum_t r_t e_t(x)\rho_1(T-t\mid x)}+\frac{\sigma^2(0,x,s)}{\sum_t r_t(1-e_t(x))\rho_0(T-t\mid x)}+\frac{\overline{\sigma}^2(1,x)}{\sum_t r_t e_t(x)}+\frac{\overline{\sigma}^2(0,x)}{\sum_t r_t(1-e_t(x))}.$$

For simplicity, we can omit the dependence on $x$ and $s$ values and focus instead on the following:

$$\min_{\delta\leq e_t\leq 1-\delta}\frac{\sigma_1^2}{\sum_t r_t e_t\rho_1(T-t)}+\frac{\sigma_0^2}{\sum_t r_t(1-e_t)\rho_0(T-t)}+\frac{\overline{\sigma}_1^2}{\sum_t r_t e_t}+\frac{\overline{\sigma}_0^2}{\sum_t r_t(1-e_t)}.$$

We want to compute how many stages it takes to obtain a nontrivial solution for the adaptive experiment. The last $D^*$ stages will just come with delay; the first $T-(D^*+1)$ stages will have $\rho_1(T-t)=\rho_{1,\infty}$ and $\rho_0(T-t)=\rho_{0,\infty}$. We can condense the first $T-(D^*+1)$ variables as a single one by representing it as an average $e$:

$$\min_{\delta\leq e_t\leq 1-\delta}\frac{\sigma_1^2}{\sum_{t\leq T-D^*-1}r_t e\rho_{1,\infty}+\sum_{t>T-D^*-1}r_t e_t\rho_1(T-t)}$$

$$+\frac{\sigma_0^2}{\sum_{t\leq T-D^*-1}r_t(1-e)\rho_{0,\infty}+\sum_{t>T-D^*-1}r_t(1-e_t)\rho_0(T-t)}$$

$$+\frac{\overline{\sigma}_1^2}{\sum_{t\leq T-D^*-1}r_t e+\sum_{t>T-D^*-1}r_t e_t}+\frac{\overline{\sigma}_0^2}{\sum_{t\leq T-D^*-1}r_t(1-e)+\sum_{t>T-D^*-1}r_t(1-e_t)}.$$

We want to introduce a solution for $e$ that is bounded away from the boundary. By KKT conditions, for the optimal solution variable $e^\star$, we have

$$\frac{\partial\mathcal{L}(e;e_t)}{\partial e}\bigg|_{e=e^\star}=-\frac{\sigma_1^2\sum_{t\leq T-D^*-1}r_t\rho_{1,\infty}}{\{\sum_{t\leq T-D^*-1}r_t e^\star\rho_{1,\infty}+\sum_{t>T-D^*-1}r_t e_t\rho_1(T-t)\}^2}$$

$$+\frac{\sigma_0^2\sum_{t\leq T-D^*-1}r_t\rho_{0,\infty}}{\{\sum_{t\leq T-D^*-1}r_t(1-e^\star)\rho_{0,\infty}+\sum_{t>T-D^*-1}r_t(1-e_t)\rho_0(T-t)\}^2}$$

$$-\frac{\overline{\sigma}_1^2\sum_{t\leq T-D^*-1}r_t}{\{\sum_{t\leq T-D^*-1}r_t e^\star+\sum_{t>T-D^*-1}r_t e_t\}^2}$$

$$+\frac{\overline{\sigma}_0^2\sum_{t\leq T-D^*-1}r_t}{\{\sum_{t\leq T-D^*-1}r_t(1-e^\star)+\sum_{t>T-D^*-1}r_t(1-e_t)\}^2}=0.$$

Note that the derivative function $\partial\mathcal{L}/\partial e$ is continuous and monotonically increasing in $e$. We can upper bound the minimum value of $\partial\mathcal{L}/\partial e$ and set it to negative:

$$\frac{\partial\mathcal{L}}{\partial e}\Big|_{e=\delta} \leq -\frac{(\sigma_1^2/\rho_{1,\infty}+\overline{\sigma}_1^2)\sum_{t\leq T-D^*}r_t}{\{\sum_{t\leq T-D^*}r_t\delta+\sum_{t>T-D^*}r_t\}^2} + \frac{(\sigma_0^2/\rho_{0,\infty}+\overline{\sigma}_0^2)\sum_{t\leq T-D^*}r_t}{\{\sum_{t\leq T-D^*}r_t(1-\delta)\}^2} \leq 0.$$

Denote

$$R_f = \sum_{T\leq T-D^*}r_t, \quad R_d = \sum_{T>T-D^*}r_t.$$

Here $R_f$ stands for the portion of the whole sample before and including stage $T-D^*$, and $R_d$ stands for the portion beyond stage $T-D^*$. After simplification, we obtain

$$\frac{R_f\delta+R_d}{R_f(1-\delta)} \leq \frac{\sigma_1^2/\rho_{1,\infty}+\overline{\sigma}_1^2}{\sigma_0^2/\rho_{0,\infty}+\overline{\sigma}_0^2},$$

which further simplifies to

$$R_f \geq \frac{1}{1-\delta}\cdot\frac{\sqrt{\sigma_0^2/\rho_{0,\infty}+\overline{\sigma}_0^2}}{\sqrt{\sigma_1^2/\rho_{1,\infty}+\overline{\sigma}_1^2}+\sqrt{\sigma_0^2/\rho_{0,\infty}+\overline{\sigma}_0^2}}.$$

Similarly and symmetrically, we can lower bound the maximal value of the derivative and set it to positive, which gives

$$R_f \geq \frac{1}{1-\delta}\cdot\frac{\sqrt{\sigma_1^2/\rho_{1,\infty}+\overline{\sigma}_1^2}}{\sqrt{\sigma_1^2/\rho_{1,\infty}+\overline{\sigma}_1^2}+\sqrt{\sigma_0^2/\rho_{0,\infty}+\overline{\sigma}_0^2}}.$$

Summarizing both parts, we obtain

$$R_f \geq \frac{1}{1-\delta}\cdot\frac{\max\left\{\sqrt{\sigma_0^2/\rho_{0,\infty}+\overline{\sigma}_0^2},\sqrt{\sigma_1^2/\rho_{1,\infty}+\overline{\sigma}_1^2}\right\}}{\sqrt{\sigma_1^2/\rho_{1,\infty}+\overline{\sigma}_1^2}+\sqrt{\sigma_0^2/\rho_{0,\infty}+\overline{\sigma}_0^2}}.$$

**Step 2. Convergence of the empirical objective.**

We prove that the empirical version of the objective function converges uniformly to the population program:

$$\mathcal{E}_N = \max_{e\in[\delta,1-\delta]^T}|\widehat{\mathcal{L}}(e)-\mathcal{L}(e)| = O_p(\frac{1}{\sqrt{N}}). \tag{5}$$

For the first $D^*+1$ stages, because the data collection policy is complete randomization with equal probability, the estimates for conditional variances, conditional means, and delay distribution are all $\sqrt{N}$-consistent. Noting that the denominators of $\widehat{\mathcal{L}}$ and $\mathcal{L}$ are strictly bounded away from 0 and 1, (5) can be verified.

**Step 3. Convergence of the solution.** Using the fact that $\widetilde{e}_N$ minimizes the empirical objective, we have

$$\widehat{\mathcal{L}}(\widetilde{e}_N)+\lambda_N\|\widetilde{e}_N\|_2^2 \leq \widehat{\mathcal{L}}(e^*)+\lambda_N\|e^*\|_2^2.$$

Then we have

$$\lambda_N\|\widetilde{e}_N\|_2^2-\lambda_N\|e^*\|_2^2 \leq \widehat{\mathcal{L}}(e^*)-\widehat{\mathcal{L}}(\widetilde{e}_N) \leq \mathcal{L}(e^*)-\mathcal{L}(\widetilde{e}_N)+2\mathcal{E}_N \leq 2\mathcal{E}_N.$$

where the last inequality follows that $e^*$ minimizes $\mathcal{L}(e)$.

Meanwhile, using

$$\mathcal{L}(e_N^*)+\lambda_N\|e_N^*\|_2^2 \leq \mathcal{L}(\widetilde{e}_N)+\lambda_N\|\widetilde{e}_N\|_2^2,$$

we have

$$\lambda_N\|e_N^*\|_2^2-\lambda_N\|\widetilde{e}_N\|_2^2 \leq \mathcal{L}(\widetilde{e}_N)-\mathcal{L}(e_N^*) \leq \widehat{\mathcal{L}}(\widetilde{e}_N)-\widehat{\mathcal{L}}(e_N^*)+2\mathcal{E}_N \leq 2\mathcal{E}_N.$$

To sum up, we have proved

$$\|e_N^*\|_2^2 - \frac{2\mathcal{E}_N}{\lambda_N} \leq \|\widetilde{e}_N\|_2^2 \leq \|e^*\|_2^2 + \frac{2\mathcal{E}_N}{\lambda_N}.$$

Now letting $N \to \infty$, we know that

$$\frac{2\mathcal{E}_N}{\lambda_N} = O_p\left(\frac{1}{\sqrt{\log N}}\right).$$

Hence we can conclude that

$$\|\widetilde{e}_N\|_2^2 \xrightarrow{\mathbb{P}} \|e^\star\|_2^2. \tag{6}$$

On the other hand,

$$\begin{aligned}
\mathcal{L}(e^\star) \leq &\mathcal{L}(\widetilde{e}_N) \\
\leq &\widehat{\mathcal{L}}(\widetilde{e}_N) + \lambda_N\|\widetilde{e}_N\|_2^2 - \lambda_N\|\widetilde{e}_N\|_2^2 + \mathcal{E}_N \\
\leq &\widehat{\mathcal{L}}(e_N^\star) + \lambda_N\|e_N^\star\|_2^2 - \lambda_N\|\widetilde{e}_N\|_2^2 + \mathcal{E}_N \\
\leq &\mathcal{L}(e_N^\star) + \lambda_N\|e_N^\star\|_2^2 - \lambda_N\|\widetilde{e}_N\|_2^2 + 2\mathcal{E}_N.
\end{aligned}$$

Therefore,

$$\mathcal{L}(\widetilde{e}_N) \xrightarrow{\mathbb{P}} \mathcal{L}(e^\star) \tag{7}$$

(6) states that the distance between the $\widetilde{e}_N$ and the Euclidean ball $\mathfrak{B}^\star = \{e : \|e\|_2 = \|e^\star\|_2\}$ converges to zero: $d(\widetilde{e}_N, \mathfrak{B}^\star) \xrightarrow{\mathbb{P}} 0$. (7) states that the distance between $\widetilde{e}_N$ and the minimal point set $\mathfrak{M}^\star$ converges to zero: $d(\widetilde{e}_N, \mathfrak{M}^\star) \xrightarrow{\mathbb{P}} 0$. But, $\mathfrak{B}^\star \cap \mathfrak{M}^\star = \{e^\star\}$. Therefore it must be that

$$\widetilde{e} \xrightarrow{\mathbb{P}} e^*.$$

**Step 4. Convergence of the implemented propensity scores.**

The rest of the result, $\widehat{e} \xrightarrow{\mathbb{P}} e^{**}$, is now a direct result from Step 2 and Step 3.

$\square$

## B  Synthetic case study details

In this section, we provide the true parameters generated from the real data adopted in our synthetic case study. The parameters are summarized in Tables B and 2.

Table 1: Delay mechanism from the real data

| $T-t$ | $X$ | $\rho_1(D \leq T-t|X)$ | $\rho_0(D \leq T-t|X)$ |
|---|---|---|---|
| 0 | 1 | 0.69 | 0.60 |
| 1 | 1 | 0.89 | 0.86 |
| 2 | 1 | 0.98 | 0.97 |
| 3 | 1 | 1 | 1 |
| 0 | 0 | 0.68 | 0.67 |
| 1 | 0 | 0.92 | 0.93 |
| 2 | 0 | 0.98 | 1 |
| 3 | 0 | 1 | 1 |

Table 2: Parameters generated from real data (Continuous outcome)

| $X$ | $S$ | $\tau(1,x,s)$ | $\tau(0,x,s)$ | $\sigma(1,x,s)$ | $\sigma(0,x,s)$ |
|---|---|---|---|---|---|
| 0 | 1 | 2.50 | 2.98 | 0.36 | 2.06 |
| 1 | 1 | 2.72 | 2.47 | 0.82 | 0.31 |
| 0 | 2 | 3.03 | 3.02 | 2.06 | 1.70 |
| 1 | 2 | 2.68 | 2.92 | 0.66 | 0.85 |
| 0 | 3 | 2.94 | 2.59 | 1.27 | 0.48 |
| 1 | 3 | 3.13 | 2.84 | 2.01 | 0.78 |

