# OpenReview forum: "Using Surrogates in Covariate-adjusted Response-adaptive Randomization Experiments with Delayed Outcomes"
_NeurIPS.cc/2024/Conference — NeurIPS 2024 poster_

### Official Review · Reviewer_oWF4 · 2024-07-08

**Soundness:** 3
**Presentation:** 3
**Contribution:** 2
**Rating:** 6
**Confidence:** 3

**Summary:**

Clinical trials try to achieve the highest statistical precision using the fewest number of enrolled participants. One way to do this is by assigning more participants to the {covariate, arm} combinations with the highest outcome variance. We don't know the outcome variance before running the trial, which motivates the adaptive CARA method to adaptively assign more units with high outcome variance to treatment. This paper considers the setting where we don't directly observe the outcome, but instead observe a noisy surrogate along with a delayed outcome. The paper proposes a design objective (minimizing the semiparametric efficiency bound) and an explore-then-commit style algorithm for learning then applying a covariate-adaptive randomization using surrogate outcomes. The paper shows that this algorithm attains the semiparametric efficiency bound, and provides synthetic experiments based on an HIV study comparing this algorithm to complete randomization and CARA on observed outcomes only.

**Strengths:**

I am not very familiar with real-world clinical trials. However, from what I can tell, the work is well-grounded in problems that clinical trial designers care about. The mathematical model of surrogacy and delay is simple to state and understand while capturing the important aspects of the problem. The problem is explained clearly with explicit examples.

**Weaknesses:**

The empirical results are missing error bars. These should be added to show variation in the results across many replicates. This is especially important for two reasons: (a) to support the claim that the proposed method has lower bias than complete randomization, when the two are extremely close on the plots, and (b) to understand whether the "no surrogate" method is really worse in both bias and variance than complete randomization, as suggested by figure 2. If the "no surrogate" method is actually worse than complete randomization, it would be interesting to have a comment on this in the text.

**Questions:**

I had trouble understanding the setup of the real-world study. If I understand correctly, measurements are taken every 6 months. The actual outcome Y is viral load (measured every 6 months), while the surrogate outcome is WHO stage (also measured every 6 months). I'm confused by the claim that WHO stage is available immediately, which I took to mean that the experimenters receive the WHO stage for every patient every 6 months. Does that not require the patient to visit a clinic, which might also lead to missingness? This isn't a question you need to address in your response, but if I have misunderstood something it might be worth clarifying.

Request: Neurips style is for citations to be by (Author, Year) instead of numerical. Please fix this in the next revision.

The introduction states that "we allow D_it = infty" (Line 139), but the algorithm only works with finite, known upper bound on D. It would be helpful for the authors to address the sensitivity of their results to misspecification of D*.

**Limitations:**

Yes

---

> ### Author Rebuttal · Authors · 2024-08-06
>
> Thanks for the careful review and critical comments. Below we give a point-by-point review to your comments.
>
> - *I'm confused by the claim that WHO stage is available immediately, which I took to mean that the experimenters receive the WHO stage for every patient every 6 months. Does that not require the patient to visit a clinic, which might also lead to missingness?*
>
> Re: Thanks for your question. WHO stage is a measure of the severity of HIV symptoms for the patients, and it can be collected from phone call check-in. However, clinical visits are required to measure HIV viral load. We will clarify this point in the next revision.
>
> - *Request: Neurips style is for citations to be by (Author, Year) instead of numerical. Please fix this in the next revision.*
>
> Thanks for pointing this out. We will correct the style in the next revision.
>
> - *The introduction states that "we allow $D\_{it} = \\infty$" (Line 139), but the algorithm only works with finite, known upper bound on $D$. It would be helpful for the authors to address the sensitivity of their results to the misspecification of $D^\*$.*
>
> Re: We apologize for the confusion. More rigorously, Our framework actually allows the delay to take a finite number of values, instead of having a finite upper bound. Therefore, we can also add in $\infty$ as long as there is a finite support for the rest of the values. In the next revision, we will further clarify this point.
>
>
>
> - *Weakness: The empirical results are missing error bars.*
>
> Re: Thank you for your suggestion. We have added confidence bands to the bias and standard deviation plots. See Figure 2 of the attached PDF file. From the bias plot Figure (A), we observe that the complete randomization design becomes unbiased when the sample size is over 800, while our proposed design is unbiased at a much smaller sample size. This suggests that our proposed design has a smaller finite sample bias. In Figure (B), our method demonstrates a significantly smaller standard deviation compared to the other two designs. For the "No surrogate" design (the green bands in both plots), we observe that its bias is rather significant in plot (A) as the green band fails to cover 0 when the sample size is smaller than 1600. We will further clarify this point in the next revision.
>
>
> Thanks very much again for your engagement with our paper. We will incorporate your comments in the next revision.

---

> > ### Comment · Reviewer_oWF4 · 2024-08-09
> > **Request for clarification**
> >
> > Thank you for your responses. I have two remaining questions:
> >
> > * $D^* = \infty$: Your response suggests that D* can be infinite, as long as the delay takes only a finite number of values. Line 183 says that the algorithm requires at least T=(2D* + 2) steps. If D* is infinite, how would the algorithm ever move to Stage 2, which happens after D*+2 steps?
> >
> > * Error bars. Thank you for adding these. My read of Figure 2 in your pdf is that complete randomization and the proposed method have identical bias, given the width of the confidence intervals. In other words, it's not possible to say that the bias of one method is better or worse than the other. I'm not sure what you mean by the claim "the complete randomization design becomes unbiased when the sample size is over 800, while our proposed design is unbiased at a much smaller sample size." I suspect this claim might be because the _confidence interval_ for the bias crosses zero slightly earlier for the proposed method than the completely randomized design. However, I think that interpretation of the plot is incorrect. What we actually care about is the mean of the bias, which is essentially identical for the two methods. I think the claims about smaller bias are not justified by the plots. The claims about variance (Fig 2B) are well-supported.

---

> ### Author Response · Authors · 2024-08-12
>
> Thanks for your response.
>
> - Thank you for pointing this out. Apologize that we did not state the definition of $D^\star$ clearly in the paper. For the definition of $D^\star$, it is the upper bound for the finite values that delay can take. In the equation following Line 134, we briefly stated the values that $D$ can take. Moreover, Line 182 of the paper is not correct - it should be stated as $\mathcal{D} = \\{0,1,\dots,D^\star\\}\cup\\{\infty\\}$.
>
>    From a high level, when the delay takes its value on finite support (no need to be finitely bounded), we can more easily estimate the delay distribution by taking empirical means and avoid unnecessary technicality in our theoretical discussion.
>
> - Apologize for the ambiguity in our wording. By "unbiased" we want to make the claim of asymptotic unbiasedness of the three approaches. We fully agree that all three methods are asymptotically unbiased or asymptotically consistent for the truth, or in your words, "the mean of the bias is identical". Our claim wants to highlight a bit on the finite sample performance of the estimators in this example; or, how fast can these bias converge to zero. The confidence band suggests the "No surrogate" approach converges slightly slower, which makes sense because the surrogate information is not used in the implementation to mitigate the missingness in primary outcomes. The proposed method and complete randomization behave quite similar. Therefore, while asymptotically all three methods are unbiased, incorporating surrogates can slightly improve the finite sample convergence rate of the final estimator.
>
> Hope the above clarification resolves your concerns. We will also incorporate the updates in the next revision.

---

> > ### Comment · Reviewer_oWF4 · 2024-08-12
> >
> > Thanks for the response.
> >
> > ## Definition of $D^*$
> > Okay, interesting -- so I'm guessing you can learn the delay mechanism in $D^*$ steps by assuming that every unit which was not observed after $D^*$ steps has infinite delay. I believe you that this makes the theory easier, although it doesn't align with common delay distributions like the exponential. I suppose another approach could be to assume a parametric form for the delay and fit that on the data from Stage 1.
> >
> > You don't need to respond to this; I'm just sharing some thoughts.
> >
> > ## Bias in rebuttal Figure 2A.
> >
> > I think we're still not understanding one another. I also agree that all the methods are asymptotically unbiased, but this is not what I meant when I referred to the mean of the bias. Instead, I meant that in Figure 2A, the blue squares (complete randomization) and orange triangles (proposed method) are plotted exactly on top of each other. I expect you computed those points by running finite-sample simulations with N="sample size" observations, computing the actual bias for each simulation, and averaging those biases. Therefore it looks to me like the finite-sample mean bias is the same between complete randomization and the proposed method.
> >
> > I want to make sure that you're not claiming the proposed method has superior bias properties to complete randomization on the basis of Figure 2A. The proposed method has a _slightly_ wider confidence band on the bias than complete randomzation, but adding variance to the bias cannot be a basis for claiming that the bias is improved.
> >
> > My final request: Can you tell me how you would rewrite the sentence on line 292-293 in the paper, given the confidence intervals you have presented in this rebuttal? (Specifically, the line "Figure 2(A) suggests...estimating the ATE.")

---

> ### Author Response · Authors · 2024-08-12
>
> Thanks for the fast response.
>
> - You are right about the intuition. The idea is just to learn the observed delays and do a subtraction to learn the permanent missing (or censoring) probability. This gives an easier theoretical discussion and also provides a fully nonparametric framework. Putting a parametric form works too and is also a great idea to extend the framework to allow infinite delays. On the algorithm side, it does not change the general pipeline too much except for the delay estimation part. We also need to modify the theoretical argument accordingly.
>
> - Oh we see your point. Yes, based on the confidence band, there is no significant difference in the mean of the bias. **We do not claim that the mean of bias of the proposed method has superior properties.** Our point is that the proposed design and estimator leads to a reduction of variance while maintaining the asymptotical unbiasedness property, and we show this both theoretically and empirically. This is the main message we want to convey, combining both figures.
>
> Re your request, we hope to modify the wording as follows:
>
> > Figure (A) suggests that the point estimators from all three designs have a vanishing bias as the sample size grows, validating the asymptotic unbiasedness of all strategies. Nevertheless, in terms of variance, Figure (B) demonstrates that our proposed design yields smaller standard deviations and thus higher estimation efficiency for estimating the ATE.
>
> Thanks again for these questions. All the points are well taken and will be incorporated in our next round of discussion.

---

> > ### Comment · Reviewer_oWF4 · 2024-08-12
> >
> > Great, thank you for the discussion.
> >
> > The authors have addressed my concerns, and I will raise my score to a 6.

---

### Official Review · Reviewer_GAcb · 2024-07-09

**Soundness:** 3
**Presentation:** 1
**Contribution:** 3
**Rating:** 6
**Confidence:** 3

**Summary:**

This article studies how to use interminate surrogate outcomes to estimate causal effects when the primary outcome is delayed in clinical trials. The author first proposes a novel Covariate-adjusted Response-adaptive (CARA)  design that supports efficient estimation of ATE using both surrogate and primary outcomes.  The authors prove the semiparametric efficiency bound for their ATE estimator and demonstrate the proposed design's efficiency through theoretical analyses and a synthetic HIV study.

**Strengths:**

1. The delay outcome problem studied in this paper is well-motivated. The covariate-adjusted response-adaptive (CARA)  design allows us to modify the treatment allocation mechanism over time, providing an interesting scenario to address the problem.
2. The authors consider improving efficiency in both design and estimation. They show that the proposed design strategy converges to an oracle design and proposes a semiparametric efficient estimator of the ATE.
3. The paper includes a synthetic HIV trial to illustrate the practical application and efficiency gains of the proposed design. The study demonstrates that the design reduces the standard deviation in estimating the ATE compared to other methods.

**Weaknesses:**

The main weakness of this article is that the presentation is too messy and lacks explanations in multiple places.
1. In section 3, from line 159 to 179, the author introduces the efficient influence function (EIF) and variance bound in estimating $\tau$. However, it seems like the main motivation of CARA is that the expectation of the delay outcomes and the delay mechanisms are unknown.   It is unclear why EIF and variance bound are presented at the beginning of section 3.

2. Section 4 is also presented in a poor way. The authors go through the design steps with dense notation. The authors should provide more explanation of the steps taken in the design. The current Section 4 only shows that the design allows us to estimate the unknown quantity. But it is unclear how the design improves efficiency.

**Questions:**

N.A.

**Limitations:**

N.A.

---

> ### Author Rebuttal · Authors · 2024-08-06
>
> Thanks for your careful review and critical comments. Below we add a point-by-point response to the weaknesses you listed.
>
> - *The presentation is too messy and lacks explanations in multiple places. It is unclear why EIF and variance bound are presented at the beginning of section 3.*
>
> Re: We apologize for the confusion. The semiparametric efficiency bound is the building block for our adaptive design proposal, so we use the majority of Section 3 to present it. We agree that we should try to include more intuition on why we introduce the EIF and variance bound before diving into the technical details. In the next revision, we plan to add the following explanation in Section 3:
>
> > The idea of adaptation is to adaptively collect data and adjust the allocation probability across multiple experiment stages to minimize the variance of a certain estimator for ATE. One question is: what estimator should be used for treatment effect evaluation at the end of the study? For ATE estimation, one popular choice is the semiparametric efficient estimator, whose variance matches the efficiency lower bound across all semiparametric estimators. Therefore, it is a natural idea to set the semiparametric efficiency bound as the design objective and to optimize this objective with adaptive treatment allocation. As the building block, we first present the EIF and semiparametric efficiency bound in Theorem 1 to serve as the building block for our design strategy.
>
>
>
> - *Section 4 is also presented in a poor way. The authors go through the design
> steps with dense notation. The authors should provide more explanation of the
> steps taken in the design.*
>
> Re: We apologize for the confusion. In the next revision, we will try to improve the presentation of Section 4. Here is a plan for modifying Section 4:
>
>    1. We want to add more discussion on the intuition. From a high-level perspective, our proposed design consists of four steps. The first step is from Stage $1$ to $D^\star+1$, a parameter exploration step that learns the nuisance parameters such as the delay distribution, outcome function, etc. The second step is the policy optimization step after Stage $D^\star + 1$, where the experimenter uses the estimated parameter information to compute the optimal treatment allocation strategy. The third step is the policy calibration step from Stage $D^\star + 1$ to $T$, where we calibrate the treatment probability to match the optimal treatment allocation strategy computed from the policy optimization step. The final step is constructing point and variance estimators based on the collected data.
>
>    2. We can also simplify the notation by introducing some high-level explanations for the quantity estimation part. The nuisance parameters such as outcome regressions $\hat{\tau}^{(t)}(a,x,s)$ and $\hat{\tau}^{(t)}(a,x)$ as well delay distribution $\hat{\rho}_a^{(t)}(0 | x)$ can be obtained by taking a stratified average over each combination of the treatment and covariate levels. Also, we can deliver the expression of the point estimator $\hat{\tau}$ and the variance estimator $\hat{\mathbb{V}}$ to the Appendix.
>
> Thanks very much again for your engagement with our paper. We will incorporate your comments in the next revision.

---

### Official Review · Reviewer_AMr4 · 2024-07-12

**Soundness:** 3
**Presentation:** 3
**Contribution:** 2
**Rating:** 6
**Confidence:** 3

**Summary:**

This paper introduces an approach to optimizing treatment allocation for variance reduction, in the context of adaptive clinical trials.  The particular setting is one where there are delays in observing outcomes. These delays that are independent of the outcomes themselves, but they impact efficiency of estimation.  Moreover, there are short-term surrogate variables that can be used to improve estimation efficiency.  In this setting, this paper proposes to estimate the variance of different treatment allocation schemes from an initial stage of experimentation, and then use those estimates to choose an optimal treatment allocation scheme for the remainder of experimentation.

---------------
REBUTTAL UPDATE: I have updated my score from 5->6

**Strengths:**

Overall, I found the setup of the problem to be original, and while I did not have time to review the proofs in depth, the results seems correct based on my knowledge of semiparametrics.  The theoretical results are fairly thorough: For instance, I appreciated Theorem 3, which speaks to the convergence of the chosen strategy to the optimal one.  In terms of clarity, the presentation of the technical details is similarly thorough, and while a bit dense and harder to skim, I found it straightforward to follow as written.  I also appreciated the experimental results, which I found to be reasonably compelling.

**Weaknesses:**

I noted a few areas of the paper that appeared a bit weaker, or at least warrant some clarification.  I'll number my points to make it easier to respond during the rebuttal period, and put them in rough order of priority.

(W1) Significance / Novelty of Theorem 1: I understood Theorem 1 to be a necessary pre-cursor to the proposed adaptive strategy, but not the main contribution of the paper, so this concern is not as pressing as it could be.  However, it seems that Theorem 1 is essentially the same as Theorem 2.1 of Kallus & Mao 2020 [1], with the minor complication that different observations appear at different times, which leads to some additional notation to ensure the correct $e_{t}(X)$ is used in the relevant denominators.  Since the paper cites [1] (as [23] in the paper), it may be worth clarifying the similarity in the main text, and framing the contributions appropriately (e.g., my read is that Theorem 2 and 3 are the novel theoretical thrust of the paper).  I'm open to clarifications / corrections on this front.

[1] https://arxiv.org/abs/2003.12408

(W2) Significance of the proposed problem setting: From a practical perspective, I struggle to think of a setting where we would expect the causal graph in Figure 1 to hold, where the delay itself provides no information on the outcome $Y$, i.e., delays are not informative. E.g., in the HIV application, the outcome is viral load, and delay is in coming in for a visit (Lines 274-275).  It seems to be a classic case where delay could be caused by more severe illness, giving us some indication of $Y$.

(W3) I found it slightly hard to parse the optimality claim (Theorems 2 and 3 and subsequent discussion).  It's clear that the estimated treatment allocation converges in probability to the optimal treatment allocation, but does that imply that this approach yields the optimal variance?  Could there be other ways to estimate the treatment allocation that also converges to the optimal one, but has better variance properties along the way, or is that somehow ruled out?

(W4) Overall, I would have preferred to see more discussion of intuition and insights, rather than just going through the technical steps to derive the results.  It seemed that some parts could be made shorter to make space for this, e.g., Section 4 could be trimmed down a bit (e.g., define $\hat{\tau}^t(a, x, s), \hat{\tau}^t(a, x)$ and then use that notation in both Stage 1 and Stage 2 to $D^*+1$, or even just give the high-level comment that you estimate via empirical counts among the population where $Y$ is observed.)

In terms of missing intuition, I still lack some intuition for the "signal" being used in the optimization of the treatment allocation, and why we should expect it to vary across time periods.  I suppose the variance reduction comes from some understanding of which combinations of $X, A$ are likely to have missing data by the end due to delay, and so should be prioritized earlier in the treatment allocation process?

(W5) There is a slight mismatch between the discussion in the introduction and the actual setting where the algorithm can be applied, particularly the restriction that the delay must be upper-bounded (see 181-182).  In contrast, Line 75-76 refers to the contribution "optimize the statistical efficiency in the presence of temporary or permanent missingness", where I would have interpreted "permanent missingness" to be $D = \infty$.

(W6) Interaction (or lack thereof) between delay and surrogates: This is a subjective / aesthetic point, so I place it last, but I was expecting more of a nuanced interaction between the delay and the surrogates.  In particular, the problem seems to factorize cleanly into (a) deriving an efficient estimator and variance lower-bound for any treatment allocation, (b) optimizing the allocation in a backward-looking period of data, and then (c) applying that allocation to the next period of data.  It's not clear the role of surrogates in this recipe, other than the influence they have on (a), because observing surrogates gives you no insight into future delays and therefore the optimal allocation going forward.

Minor nits:
* Algorithm 1 mentions "Problem B" on line 7, but I think that's really just Equation (2), Page 6, line 199.

**Questions:**

1. (W1) Could you clarify the similarity/difference of Theorem 1 to Theorem 2.1 of Kallus & Mao 2020?  It's fine if they are essentially the same, in my view, b/c the main contribution of the paper seems to lie in Theorems 2 & 3, but I just wanted to get some clarification in case I'm missing something.
2. (W2) Is there a real-world scenario you have in mind where you would expect this causal structure to hold (i.e., delays are not informative of outcomes)?
3. (W3) Could you clarify the optimality claim?  Is it just that the allocation converges in probability to the optimal allocation?  Or is there a stronger claim being made about the optimality of the procedure which uses the estimated allocation?
4. (W4) Is my intuition correct, for where the variance reduction is coming from?  Or do you have other intuition on this point?

The authors are welcome to react to my points (W5) and (W6), but I don't think they need to do so, and they factored less into my score.

---

> ### Author Rebuttal · Authors · 2024-08-06
>
> Thanks for your careful review and critical comments. Below we add a point-by-point response.
>
> - *(W1)*: We agree that our efficiency bound may share some similarities with [1], but also hope to emphasize that deriving a neat decomposition of the tangent space is not straightforward when data come in multi-stages with delays (Step 2 in the proof of Theorem 1). This step is crucial for score function projection and finding the efficient influence function and requires more careful treatment in our setting than [1].
>
> On the other hand, as you have correctly pointed out, this result is the building block for the proposed adaptive design strategy and motivates point and variance estimators for ATE. Grounded by Theorem 1, Theorem 2 and 3 further provide guarantees for the optimality of design and validity of inference. We will further clarify these points in the next revision of our manuscript.
>
> - *(W2)*: In HIV trials, delays can occur for many reasons. For instance, viral load is typically measured every six months due to the protocol, which are independent of outcomes. Also, factors like economic status or distance to clinics are confounders for delay and outcome. Once adjusted, these delays become non-informative. Meanwhile, delays due to severity of HIV, as you mentioned, are informative for the outcome. In reality, the interplay among these factors is complex. In our simulation, we simplify the setup by considering a hypothetical setting where delays are due to protocol or structural factors.
>
>   In some cases, the delay is conditionally independent of the outcome. For example, if we are testing the effect of a vitamin supplement on blood vitamin levels and ask patients for regular check-ins, the delay is less likely to depend on the outcome since it does not cause severe symptoms like HIV. In general, there is a complicated delay-outcome interaction. We leave the extension as future work.
>
> - *(W3)*: Apologize for the confusion. The optimal allocation means the sequence of propensity scores that gives the minimal asymptotic variance matching the efficiency bound. So optimal allocation implies optimal variance. Nevertheless, the optimal treatment allocation that achieves the efficiency bound might not be unique, and one can add additional constraints or penalization (say $\ell\_2$, entropy, and so on). We will clarify this point in the next revision.
>
> - *(W4)*: Thanks for the suggestion. In the next revision, we will incorporate your suggestions to improve the presentation. Here is a modification plan:
>
>    1. More discussion on the intuition. Our proposal has four steps. Step 1 is from Stage $1$ to $D^\star+1$, an exploration step that learns parameters like the delay distribution, outcome model, etc. Step 2 is the policy optimization step after Stage $D^\star + 1$, to compute the optimal allocation. Step 3 is policy calibration from Stage $D^\star + 1$ to $T$, where we calibrate the allocation to match the optimal strategy. Step 4 is constructing point and variance estimators based on the data.
>
>    2. Simplify the notation. The elements such as outcome models $\hat{\tau}^{(t)}(a,x,s)$ and $\hat{\tau}^{(t)}(a,x)$, and delay distribution $\hat{\rho}_a^{(t)}(0 | x)$ can be obtained by taking a stratified average over each combination of the $(A,X,S)$ levels. Also, we will defer the form of $\hat{\tau}$ and $\hat{\mathbb{V}}$ to the Appendix.
>
>    Moreover, your intuition behind variance reduction is correct. In our setting, the missing probability is different across stages and combinations of $(X, A)$ levels. We need enough data in the early stage to learn the mean and variance across arms, especially for those that are more likely to delay, so that we can optimize variance in later stages. As a toy example, consider a two-stage example, with the same $n_t$ for each stage.  We omit surrogates here for simplicity. The bound in Theorem 1 becomes:
> \begin{align}
>     & \mathbb{V}\_{SPEB}
>     =\mathbb{E}[\big(\tau(1,X) -\tau(0,X) - \tau\big)^2]\\\\
>     & + 2\mathbb{E}\Big[\frac{\sigma\_1^2(X)}{ e\_1(X) \rho_1(1|X)}  + \frac{\sigma\_0^2(X)}{(1-e\_1(X)) \rho_0(1|X)}\Big]\\\\
>     & + 2\mathbb{E}\Big[\frac{\sigma\_1^2(X)}{ e\_2(X) \rho_1(0|X)}  + \frac{\sigma\_0^2(X)}{(1-e\_2(X)) \rho_0(0|X)}\Big].
> \end{align}
>    The first stage is exploration for estimating the nuisance. We set $e_1(X) = 1/2$ to guarantee each $(x,a)$ level has enough sample. For the second stage, we minimize $\mathbb{V}_{SPEB}$ over $e_2(X)$, which gives
>    \begin{align}
>        e_2^\star(x) = \frac{\sigma_1(x)/\sqrt{\rho_1(0\mid x)}}{\sigma_1(x)/\sqrt{\rho_1(0\mid x)}
>        +
>        \sigma_0(x)/\sqrt{\rho_0(0\mid x)}}
>    \end{align}
>    Therefore, the allocation depends on $\sigma_a(x)/\sqrt{\rho_a(0\mid x)}$. If some $(x,a)$ levels lead to larger variance and fewer delays, the design will sample more from those levels to facilitate variance reduction.
>
>   - *(W5)*: We apologize for the confusion. More rigorously, our framework allows the delay to take a finite number of values, instead of having a finite upper bound. Therefore, we can also add in $\infty$ as long as there is a finite support for the rest of the values. In the next revision, we will further clarify this point.
>
>    - *(W6)*: One role of surrogates here is, as you have pointed out, to facilitate the derivation of the point estimator variance bound, which suggests that incorporating surrogates reduces variance (Remark 1). Another role is that it helps with policy optimization in the early stages when many primary outcomes are missing. It is interesting to think about the interplay between delay and surrogates. As a special setup, Kallus \& Mao (2020) discussed missingness that depends on surrogates. We believe such an extension also works in our case and will leave this as a future work.
>
>   - Minor nits: will correct in the next revision.
>
> Thanks again for the insightful comments! We will incorporate these points in the next revision of our manuscript.

---

> ### Comment · Reviewer_AMr4 · 2024-08-07
>
> Thank you for the thoughtful response, a quick point-by-point of my reactions.  Overall I'm positive on the paper, and I'm going to raise my score to a 6.
>
> The one point I'm still fuzzy on is (W3):
>
> Do you need any other conditions to say something about the asymptotic variance being optimal?  To clarify my question with an example:  If I want to estimate $E[g(X)]$ for some deterministic function g, then obviously the empirical mean $E_n[g(X)]$ has optimal variance, but it's not usually the case that the plug-in $E_n[\hat{g}(X)]$ has optimal variance, even if $\hat{g}$ is consistent/converges in probability to $g$.  Similarly, you're saying that the allocation converges in probability to the optimal one, and that if you had the optimal allocation, it would minimize variance, but I struggle to see how that's different from my example, where I could say "$\hat{g}$ converges to $g$, and using $g$ gives optimal variance, therefore my procedure is also optimal" but obviously that's incorrect.  That's why I was wondering if there are other conditions, e.g., in my example above you might require that the stronger condition that $\hat{g}$ converges to $g$ at $o_p(n^{-1/2))$ rates so it doesn't contribute to the asymptotic variance, etc.
>
> For the remaining points, thank you for the clarifications, my quick takes:
> * (W1) That seems reasonable enough as a distinction, and I would clarify that point in the paper, as you state.
> * (W2) Fair enough, I would suggest some discussion in the paper as space permits, to at least highlight reasons why the assumption might be considered a strong one.
> * (W4) I like the example!  Would be nice to include somewhere for intuition, if space permits.
> * (W5-6) Makes sense, understood, good clarifications all around.

---

> > ### Author Response · Authors · 2024-08-08
> >
> > Thanks very much for your quick response and for raising the score!
> >
> > Your concern regarding the estimation problem for $E[g(X)]$ is correct for a general $X$. But in our case, we take a simplification and consider just **discrete** $X$ with finite support $\mathcal{X}$ (first paragraph of Section 4) to avoid unnecessary technical complexity. To see why this makes things go through, suppose we have $|\hat{g}(x) - g(x)| \to 0$ for every $x\in\mathcal{X}$. We can compute
> > \begin{align}
> >  & \frac{1}{n}\sum_{i=1}^n \hat{g}(X_i) \\\\
> > & = \frac{1}{n}\sum_{x\in\mathcal{X}}\sum_{i=1}^n \hat{g}(x) 1(X_i = x) \\\\
> > & = \frac{1}{n}\sum_{x\in\mathcal{X}}\sum_{i=1}^n \\{{g}(x) + o_p(1)\\} 1(X_i = x) \\\\
> > & = \sum_{x\in\mathcal{X}}g(x)p(x) + o_p(1)\\\\
> > & = E(g(X)) + o_p(1).
> > \end{align}
> > Therefore, when the support of $X$ is finite, the technical part is greatly simplified. Nevertheless, for continuous $X$, we will need more conditions as you suggested, such as root-$n$ convergence of the nuisance, and more steps to achieve the optimal variance such as sample splitting.
> >
> > We hope this short discussion clarifies your concern and will include all the points in the next revision.

---

> > > ### Comment · Reviewer_AMr4 · 2024-08-09
> > >
> > > I think again you're focused on consistency while I'm asking about something else - the implications for the asymptotic variance.  Continuing in this simple example, consider discrete X, as you do.  Suppose that $|g(x) - \hat{g}(x)| \rightarrow 0$ in probability, but that for each value of $x$ the convergence is at $O_p(\sqrt{n})$ rates, i.e., $ \sqrt{n} (g(X) - \hat{g}(X)) \rightarrow N(0, \sigma^2)$
> > >
> > > Then if you ask about $\hat{\theta} := E\_n[\hat{g}(X)]$ and $\theta\_0 := E[g(X)]$, the analysis I was looking for is something like
> > > $$\sqrt{n} (\hat{\theta} - \theta) \rightarrow N(0, \sigma\_\theta^2),$$ and showing that the variance term is minimal.  Here there are two sources of variation, because $$\sqrt{n}(\hat{\theta} - \theta) = \sqrt{n}(E\_n[\hat{g}(X)] - E\_n[g(X)]) + \sqrt{n}(E\_n[g(X)] - E[g(X)])$$  Then the second term is what you're talking about:  It has minimal variance, because you're using the right $g(X)$.  But the first term still contributes to the variance.  Feel free to correct me if I'm thinking about this incorrectly.

---

> ### Author Response · Authors · 2024-08-12
>
> Got it. We agree with you about the variance claim - if we have a general $\hat{g}$ there is an additional layer of variance due to the estimation of $g$. Nevertheless, there is some specialty in the IPW and AIPW cases. Here the propensity score and the outcome models are nuisance parameters. For the outcome model, we can achieve $O(N^{-1/2})$ convergence because of the bounded moment condition on the potential outcomes and discrete covariates. Also for the delay distribution $\rho(t\mid X)$, we can achieve $O(N^{-1/2})$ because the delay indicators are bounded, and the covariate and delay have finite support. Then for the propensity score modeling, we only require $o_p(1)$ convergence. The proof is similar to the techniques adopted in double machine learning literature. To see this, consider the following math in the classical AIPW case:
> \begin{align}
> &\frac{1}{n}\sum_{i} \frac{(Y_i - \hat{\mu}(X_i)) A_i}{\hat{e}(X_i)} + \frac{1}{n}\sum_{i} \hat{\mu}(X_i)-\mu\\\\
> =&\frac{1}{n}\sum_{i} \frac{{e}(X_i)}{\hat{e}(X_i)} \frac{(Y_i - {\mu}(X_i)) A_i}{{e}(X_i)} + \frac{1}{n}\sum_{i}\mu(X_i) - \mu \tag{1}\\\\
> +&\frac{1}{n}\sum_{i} \\{\hat{\mu}(X_i) - \mu(X_i) \\}\\{ 1 - \frac{A_i}{\hat{e}(X_i)}\\}. \tag{2} \\\\
> \end{align}
>
> For (1), we show this part gives the asymptotic minimal variance. We have
> \begin{align}
> &\frac{1}{n}\sum_{i} \frac{{e}(X_i)}{\hat{e}(X_i)} \frac{(Y_i - {\mu}(X_i)) A_i}{{e}(X_i)} + \frac{1}{n}\sum_{i}\mu(X_i) - \mu \\\\
> =& \frac{1}{n}\sum_{i} (1 + o_p(1)) \frac{(Y_i - {\mu}(X_i)) A_i}{{e}(X_i)} + \frac{1}{n}\sum_{i}\mu(X_i) - \mu \\\\
> \asymp & \frac{1}{n}\sum_{i} \frac{(Y_i - {\mu}(X_i)) A_i}{{e}(X_i)} + \frac{1}{n}\sum_{i}\mu(X_i) - \mu + o_p(n^{-1/2}).
> \end{align}
>
> For (2), we show it's of small order $o(n^{-1/2})$.
> \begin{align}
> &\frac{1}{n}\sum_{i} \\{\hat{\mu}(X_i) - \mu(X_i) \\}\\{ 1 - \frac{A_i}{\hat{e}(X_i)}\\}\\\\
> = & \sum_{x\in\mathcal{X}}\frac{\hat{\mu}(x) - \mu(x) }{\hat{e}(x)}\frac{1}{n}\sum_{i} \\{A_i - \hat{e}(x)\\}1\\{X_i = x\\}\\\\
> = & \sum_{x\in\mathcal{X}}\frac{\hat{\mu}(x) - \mu(x) }{\hat{e}(x)} \cdot \frac{1}{n}\sum_{i} \\{A_i - {e}(x)\\}1\\{X_i = x\\}
> +\sum_{x\in\mathcal{X}}\frac{(\hat{\mu}(x) - \mu(x))(e(x) - \hat{e}(x)) }{\hat{e}(x)}\frac{1}{n}\sum_{i}1\\{X_i = x\\}\\\\
> = & \sum_{x\in\mathcal{X}} \frac{O_p(n^{-1/2})}{e(x) + o_p(1)} \cdot O_p(n^{-1/2})
> +\sum_{x\in\mathcal{X}} \frac{O_p(n^{-1/2}) \cdot o_p(1)}{e(x) + o_p(1)} \cdot \\{p(x) + O_p(n^{-1/2})\\}\\\\
> = & o_p(n^{-1/2}).
> \end{align}
>
> Therefore, we can see that (1) plays the dominant role in the asymptotic regime and gives minimal variance. Again, we highlight that the specialty is due to we can estimate the outcome model in square root order so we do not need this order for the propensity score models.
>
> Hope this clarifies your concern regarding the variance part.

---

### Official Review · Reviewer_Xjru · 2024-07-15

**Soundness:** 3
**Presentation:** 4
**Contribution:** 3
**Rating:** 7
**Confidence:** 4

**Summary:**

This paper addresses the problem of covariate adaptive experimental design in the presence of time delayed outcomes. More specifically, we assume that participants are enrolled in waves, and the probability of treatment is given conditional on available user covariates. The target estimand is the average treatment effect for a long term/delayed outcome. To circumvent the issue of covariate adaptive randomization in the absence of observed outcomes, the authors consider a surrogate metrics. To derive the assignment procedure, the authors derive the influence function and semi-parametric efficiency bound, and then derive an objective based on them. A small number of empirical evaluations is provided which show the improvement in variance of the proposed estimator over complete randomization and approaches which do not incorporate surrogate information.

**Strengths:**

This is an interesting addition to the literature on adaptive experimentation, and covariate adaptive experimentation. The authors do a nice job of clearly describing the task, motivating and deriving the influence function and semiparametric efficiency bound, and introducing a relatively simple algorithm for optimizing the bound. The proofs, to my reading, are correct and well presented. The problem is of clear practical importance, the authors motivate their approach through drug trials, but there are also a number of applications in both social scientific and industrial settings where the problem of delayed and long term outcomes arises and adaptive design is desirable. The authors also do a nice job of clearly walking through each step of the proposed algorithm and describing it's function and intuition.

**Weaknesses:**

The paper focuses on asymptotic results, which is sensible and provides good empirical performance. However, it would be useful to have finite sample analysis as well, since many of the applications of experimental design are sample-starved. It would have also been nice to have seen a slightly larger set of empirical results. The authors provide a small demonstration, but ideally the behavior of the propose algorithm is more rigorously evaluated empirically.

**Questions:**

It wasn't clear to me what is being assumed about the size of each wave of participants. I assume that the authors are assuming some conditions on the number of arriving participants? Also that these participants are arriving i.i.d. over time?  Does the proposed procedure give any guidance to the setting where the experimenter can choose the number of participants to enroll at each stage?

**Limitations:**

Yes. See above for questions.

---

> ### Author Rebuttal · Authors · 2024-08-06
>
> Thanks so much for your careful review. We add a point-by-point response to your questions below.
>
> - *It would be useful to have a finite sample analysis.*
>
> Re: Thanks for the suggestion. We pursue asymptotic analysis to establish the distributional convergence and construct confidence intervals based on normal distribution approximations. In settings like A/B testing, many users are involved and asymptotic analysis suffices to help with inference. Nevertheless, we agree that in other settings such as clinical trials, sample size becomes a more important constraint which calls for a more delicate finite sample quantification. To translate the asymptotic results into a finite sample analysis, we can utilize the martingale concentration inequalities to quantify the tail of the estimates and also use Berry-Esseen bounds to obtain rates of distribution convergence. To avoid an overly lengthy paper, we hope to save these for future work.
>
>
> - *It would have also been nice to have seen a slightly larger set of empirical results.*
>
> Re: Thanks for the suggestion. In the rebuttal, we have added a new experiment to compare different tuning strategies for our design optimization program (Figure 1 in the attached pdf). Also, we have refined the presentation of the previous experiments by adding confidence bands to the curves (Figure 2 in the attached pdf).
>
> - *It wasn’t clear to me what is being assumed about the size of each wave of participants.*
>
> Re: In our setting, we assume that in each stage, the portions of participants to be recruited in each stage (the $r_t$'s) are decided a priori. This aligns with the practice in many real-world applications. For example, in clinical trials, researchers need to decide the number of patients to enroll as part of the protocol before the trial starts. Theoretically, this assumption makes it easier to discuss asymptotic regimes when $N$ goes to $\infty$.
>
> - *Also these participants are arriving i.i.d. over time?*
>
> Re: In our setting, we assumed that the **potential outcomes of the participants across stages are i.i.d., but the truly observed outcomes are not**. The i.i.d. assumption is only imposed on the potential outcomes, which can be interpreted as the patients are drawn from the same target population. Nevertheless, due to the adaptive recruitment process, the observed outcomes are decided by both the distribution of the potential outcomes and the history-dependent data collection policy and thus are not i.i.d. anymore.
>
> - *Does the proposed procedure give any guidance to the setting where the experimenter can choose the number of participants to enroll at each stage?*
>
> Re: Thanks for this insightful point. Our results can provide some guidance in the following two aspects:
>
> 1. For Stage $1$ to $D^\star$, Condition (4) in Theorem 3 provides a sufficient condition to quantify the portion of people to recruit for the first $D^\star$ stages to achieve the optimal treatment allocation strategy, which depends on the variance and the delay mechanism. Therefore, a promising strategy is to use the first stage to estimate the size of the quantity on the right-hand side of Condition (4), then adjust the portions in the following stages to satisfy the constraint.
>
> 2. For Stage $D^\star + 1$ to $T$, we apply the calibrated treatment allocation probabilities $\hat{e}\_l$. To make sure $\hat{e}\_l$ is bounded away from $0$ and $1$, we could set up a sequence of $n_s$ that satisfy the following inequality:
> $$
>     \delta \le \frac{\sum\_{s=1}^{T-D^\star} n_s \cdot \tilde{e}\_{s} - \sum_{s=1}^{D^*+1} n\_s\cdot \hat{e}\_{s}}{\sum\_{s = D^* + 2}^{T-D^*}n\_{s}} \le 1 - \delta.
> $$
> This will ensure a sufficient portion of participants in both the treatment and control groups for Stage $D^\star + 1$ to $T$.
>
>
> Thanks again for the insightful comments! We will incorporate these points in the next revision of our manuscript.

---

> > ### Comment · Reviewer_Xjru · 2024-08-14
> >
> > Thank you for your responses. Overall, I am quite positive on this paper (as my initial score indicates). I will leave my score unchanged.

---

### Official Review · Reviewer_32pT · 2024-07-17

**Soundness:** 3
**Presentation:** 3
**Contribution:** 3
**Rating:** 5
**Confidence:** 2

**Summary:**

The paper considers the covariate-adjusted response-adaptive randomization design for settings with delayed outcomes and surrogate outcomes. The paper first characterizes the efficient influence function for estimating the primary outcome under the delayed setting with surrogate outcomes. They then characterize the semiparametric efficiency bound of the estimate. Using the semi parametric efficiency bound they devise an optimization approach to construct an adaptive randomization design that optimizes the semi-parametric efficient bound. They provide a synthetic case study to demonstrate the effectiveness of their approach.

**Strengths:**

The paper overall feels well written and well organized. The problem set-up as well as the key results follow a logical flow and it is straightforward to understand how the adaptive randomization proposed in the paper is derived. The authors also provide insightful comments on the results in the paper which helped point out its main contributions compared to existing work.

The paper's contributions are interesting as their adaptive randomization design in the delayed outcome setting with surrogate outcomes is a realistic setting seen in clinical trials performed by the FDA. The synthetic case study helps highlight the potential application of their work. They also provide theoretical results that give guarantees on the quality of the estimation approach

**Weaknesses:**

The paper leaves out some details which may be obvious but are worth including for clarity. For example, the unconfoundedness and arm-dependent delay assumptions are briefly mentioned in passing, but are not formally defined even though they are used in the statements of the theorems. The authors also do not comment on the tractability of optimization problem that determines the treatment allocation probabilities. Other details omitted include details of the cross-validation set-up used to tune the regularization penalty parameter and the the details of the "No Surrogate" approach in the case study.

Discussion on the settings where the proposed method has the most benefit is also lacking. The numeric case study primarily studies the impact on sample size on different approaches for estimating the primary outcome. The results show that while there is benefit from using the proposed method, the most naive method with complete randomization also performs better than other adaptive randomization approaches. Adaptive randomization design may be more challenging to implement so a deeper study on settings that benefit the most from the proposed method would shed more light on the benefits and robustness of the proposed approach.

**Questions:**

1. In the numerics, the "No Surrogate" approach seems to perform worse than complete randomization. Is there any insight on why this adaptive approach performs worse than the complete randomization approach? Does the "No Surrogate" approach also account for delays or does it utilize an existing adaptive randomization design approach?

2. The proposed method suggests applying regularization to the oracle optimization program to deal with potential multiple optima. Does the procedure tune the regularization parameter so it produces calibrated treatment allocation probabilities with better estimation error? Does this align cross-validation set-up briefly mentioned in the paper? If so, what is the exact cross-validation set-up and if not how is the the tuning parameter chosen?

**Limitations:**

The authors have adequately addressed limitations.

---

> ### Author Rebuttal · Authors · 2024-08-06
>
> Thank you very much for your careful review and critical questions. Below, we add a point-by-point response to your questions:
>
> - _In the numerics, the "No Surrogate" approach seems to perform worse than complete randomization. Is there any insight on why this adaptive approach performs worse than the complete randomization approach? Does the "No Surrogate" approach also account for delays or does it utilize an existing adaptive randomization design approach?_
>
> Re: Sorry for the confusion. The complete randomization strategy **incorporates surrogates and adjusts for delays** in the primary outcome. The "No Surrogate" strategy **accounts for delays but does not incorporate any surrogates**. In general, these two strategies are not directly comparable, and the point we want to emphasize here is that **both strategies are inferior to the proposed CARA design**. Between complete randomization and the "No surrogate" strategy, which performs better **depends on the surrogates' quality**. If the surrogates are more predictive of the outcome, then incorporating the surrogates can greatly reduce the bias and variance, leading to an estimator that outperforms the "No Surrogate" strategy. According to the summary statistics in Table 2 in the Appendix, different levels of the WHO HIV stage lead to a significant variation in the conditional mean and variance of primary outcomes, which suggests the importance of incorporating surrogates and explains the superiority of this strategy in this case.
>
> - _The proposed method suggests applying regularization to the oracle optimization program to deal with potential multiple optima. Does the procedure tune the regularization parameter so it produces calibrated treatment allocation probabilities with better estimation error? Does this align with the cross-validation set-up briefly mentioned in the paper? If so, what is the exact cross-validation setup and if not how is the the tuning parameter chosen?_
>
> Re: Thank you for your question. We provide a comparison of various tuning parameter selections in Figure 1 of the attached pdf file. Overall, our design is not sensitive to tuning parameter choices, especially when the sample size is large. Based on Figure 1, as heuristic guidance, we recommend choosing a tuning parameter between 0 and 1 because a large tuning parameter may push the treatment allocation solutions to the boundary. As demonstrated in Figure 1, when the tuning parameter is chosen to be 10, the design is less efficient than the performance under a smaller tuning. We agree that having a cross-validation procedure for tuning parameter selection would be more interesting and systematic. We hope to explore this piece of methodological development in future work.
>
> Thanks again for your engagement with our paper. We will incorporate your comments in the next revision.

---

### Author Rebuttal · Authors · 2024-08-06

We sincerely appreciate all the constructive feedback provided by our reviewers. We have made our best efforts to respond to the questions and comments raised by our reviewers.

To answer some of the questions raised by our reviewers, we have attached a pdf file containing the following two figures:
- Figure 1: Comparison of bias and standard deviation across various tuning parameter selections.
- Figure 2: Addition of confidence bands to the bias and standard deviation comparison plot.

Thank you once again for your time and effort in reviewing our manuscript!

---

### Decision · Program_Chairs · 2024-09-25

**Decision:**

Accept (poster)

**Comment:**

I concur with the reviewers' positive assessment of the paper. In particular, I think the problem is very well motivated from practice and merits attention, and I think the paper takes a very reasonable approach and studies it very nicely. Well done! It is thus a great addition to the literature on the subject, and I am very glad to recommend acceptance. There are some edits discussed in the rebuttal that I request the authors undertake. Also, while the authors do a nice job of reviewing the landscape of the literature, they could be more transparent in connections in their technical results. In particular, as one reviewer highlights, the connection of Theorem 1 to Theorems 2.1 and 2.2 of [23] and any significant differences should be discussed upfront (you mention some non-specific technical difficulties in the rebuttal but I did not follow since the decomposition seems to just follow from the ancillarity of the design, no?). Moreover, drawing the connection between an adaptive experiment and a sequence of different experiments and discussing the recent works of Bibaut et al. "Learning the Covariance ..." and "Nonparametric Jackknife ..." would add to the (already very nice) discussion of the landscape. I fully trust that the authors can undertake all this appropriately within the preparation of a camera ready version as this mostly involves adding a couple sentences here and there.